# UniGEM: A Unified Approach to Generation and Property Prediction for Molecules

**Shikun Feng [1]\*, Yuyan Ni[2,3]\*[†], Yan Lu[4], Zhi-Ming Ma[2], Wei-Ying Ma[1], Yanyan Lan[1,5] [‡]**

[1]Institute for AI Industry Research (AIR), Tsinghua University
[2]Academy of Mathematics and Systems Science, Chinese Academy of Sciences
[3]University of Chinese Academy of Sciences
[4]Department of Computer Science and Technology, Tsinghua University
[5]Beijing Academy of Artificial Intelligence

## Abstract

Molecular generation and molecular property prediction are both crucial for drug discovery, but they are often developed independently. Inspired by recent studies, which demonstrate that diffusion model, a prominent generative approach, can learn meaningful data representations that enhance predictive tasks, we explore the potential for developing a unified generative model in the molecular domain that effectively addresses both molecular generation and property prediction tasks. However, the integration of these tasks is challenging due to inherent inconsistencies, making simple multi-task learning ineffective. To address this, we propose UniGEM, the first diffusion-based unified model to successfully integrate molecular generation and property prediction, delivering superior performance in both tasks. Our key innovation lies in a novel two-phase generative process, where predictive tasks are activated in the later stages, after the molecular scaffold is formed. We further enhance task balance through innovative training strategies. Rigorous theoretical analysis and comprehensive experiments demonstrate our significant improvements in both tasks. The principles behind UniGEM hold promise for broader applications, including natural language processing and computer vision.[1]

## 1 Introduction

Artificial intelligence, particularly through deep learning technologies, is advancing various applications in drug discovery. This encompasses two major tasks: molecular property prediction (Zaidi et al., 2022; Feng et al., 2023a; Ni et al., 2023; 2024) and molecule generation (Hoogeboom et al., 2022; Guan et al., 2023; Gao et al., 2024). The objective of molecular property prediction is to learn functions that accurately map molecular samples to their corresponding property labels, which can facilitate the virtual screening process. Meanwhile, molecule generation aims to estimate the underlying molecular data distribution, offering significant potential for automatic drug design. Although considerable research has been conducted in these areas, they have largely progressed independently, overlooking their intrinsic correlations.

In our opinion, the essence of these two tasks lies in molecular representations. On the one hand, the effectiveness of various molecular pre-training methods demonstrates that molecular property prediction relies on robust molecular representations as a foundation. On the other hand, molecule generation requires a deep understanding of molecular structures, enabling the creation of good representations during the generation process. Recent research findings provide support for our perspective. For instance, researchers in the computer vision field have shown that diffusion models inherently possess the ability to learn effective image representations (Chen et al., 2024; Hudson et al., 2024; Mittal et al., 2023). In the molecular domain, studies have indicated that generative pre-training can enhance molecular property prediction tasks (Liu et al., 2023; Chen et al., 2023a), particularly through data augmentation using diffusion models. However, these methods often require

---

\*Equal contribution. [†] Work was done while Yuyan Ni was a research intern at AIR. [‡] Correspondence to `lanyanyan@air.tsinghua.edu.cn`.

[1]The code is released publicly at `https://github.com/fengshikun/UniGEM`.

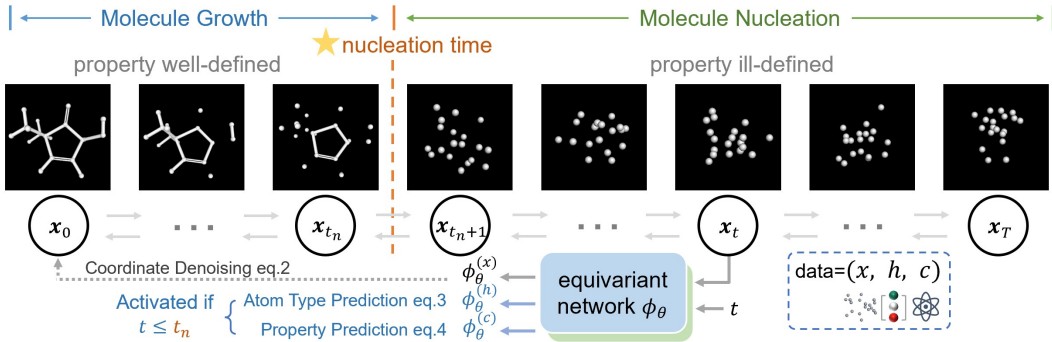

Figure 1: The two-phase generative process of UniGEM. We treat molecule generation as a two-phase problem: nucleation and growth, defining the separation time as nucleation time. Properties are only well-defined in the growth phase, so during training, property and atom type predictions are incorporated only in the growth phase.

additional fine-tuning to achieve optimal predictive performances. Additionally, while predictive tasks can guide molecule generation (Bao et al., 2022; Gao et al., 2024), it remains unclear whether they can directly enhance generative performance. Therefore, existing studies have not sufficiently elucidated the relationship between generation and prediction tasks, raising a critical question: can we develop a unified model to enhance the performance of both tasks?

A straightforward approach to combining these two tasks is to use a multi-task learning framework, where a model is trained to be simultaneously supervised by both property prediction loss and generation loss. However, as shown in Table 4 our experiments indicate that this approach (third row) significantly degrades the performance of both the generation and property prediction tasks. Even if we freeze the weights of the generation model and add a separate head for the property prediction task to maintain the generation performance (fourth row), we observe that the property prediction performance does not improve compared to training from scratch.

We attribute the suboptimal results to the inherent inconsistency between the generative and predictive tasks. In the generative task, the model operates throughout the entire diffusion process, transitioning molecules from a disordered state to a structured form. However, for predictive tasks, meaningful molecular properties can only be defined once the molecular structure has been fully established. Disordered molecules in earlier stages lack meaningful properties and can introduce errors. Consequently, merely adopting a simple combination approach results in an incorrect mapping between perturbed molecules and their properties, negatively impacting both molecule generation and property prediction. To validate the conjecture, we conduct a theoretical analysis of the mutual information between intermediate representations within the denoising network and the original data during diffusion training. According to the InfoMax principle (Linsker, 1988; Oord et al., 2018), a representation is better if the mutual information between data and the representation is larger. We theoretically demonstrate that the diffusion model implicitly maximizes the lower bound of this mutual information, suggesting the potential to unify generative and predictive tasks. However, the mutual information decreases monotonically throughout the diffusion process, approaching zero at larger time steps, indicating that representations at higher noise levels (i.e., disordered molecules) are ineffective. Thus, both intuition and theory suggest that generative and predictive tasks align only at smaller time steps when molecules remain more structured.

Based on these theoretical and empirical findings, we propose a novel two-phase generative approach to unify property prediction and generation, as shown in Figure 1. We divide the molecular generation process into two phases: the 'molecule nucleation phase' and the 'molecule growth phase'. This division is inspired by the crystal formation process in physics (De Yoreo & Vekilov, 2003), allowing us to systematically capture the complexities of molecular assembly and enhance the efficiency of generation. In the molecule nucleation phase, the molecule forms its scaffold from a completely unstructured state, after which the complete molecule is developed based on this scaffold. These phases are separated by a 'nucleation time'. We then introduce a new diffusion model to formulate these two phases. Before the nucleation time, the diffusion model progressively generates the molecular structure. After nucleation, it continues refining the structure while incorporating prediction losses into the diffusion process, optimizing both generation and prediction tasks. Unlike traditional generative models that typically perform joint diffusion of atom types and coordinates,

our approach focuses solely on the diffusion of coordinates, treating atom type as a separate prediction task. This separation is justified by the observation that atom types can often be inferred based on the determined scaffold (Bruno et al., 2011; Vaitkus et al., 2023). Specifically, prior to nucleation, the diffusion process aims to reconstruct the coordinate. Following nucleation, we incorporate predictive losses for both atom type and properties into the unified learning framework.

Our experimental results demonstrate that UniGEM achieves superior performance in both the predictive and generative tasks. Specifically, in molecule generation, UniGEM produces significantly more stable molecules, with molecular stability improving by approximately $10\%$ over the diffusion-based molecular generation model EDM (Hoogeboom et al., 2022). In terms of property prediction, our method significantly outperforms the baseline trained from scratch, even achieving accuracy comparable to some pre-training methods without introducing any additional pre-training steps and large-scale unlabeled molecular data.

To explain the improved generation performance of our unified model, we derive upper bounds on the molecular generation error for both UniGEM and traditional diffusion models. Our analysis reveals that UniGEM significantly reduces the dimensionality of diffusion data and simplifies the treatment of the discrete atom types by framing it as a prediction task, leading to lower total molecular generation errors and producing more stable and accurate molecular structures.

To the best of our knowledge, we are the first to propose an effective diffusion-based unified model that significantly enhances of the performance of both molecular generation and property prediction tasks, demonstrating a novel approach in this domain. Our treatment of molecular generation as comprising two phases, molecular nucleation and molecular growth, provides a novel paradigm that may inspire the development of more advanced molecule generation models. Furthermore, the analysis and design of how to unify generative and predictive models may also impact other AI applications, like natural language processing and computer vision.

## 2 METHOD

### 2.1 UNIGEM FRAMEWORK

UniGEM is a two-phase diffusion based generative approach for property prediction and molecular generation. For the convenience of comparing with the traditional joint diffusion approach, we adopt a notation scheme consistent with the E(3) Equivariant Diffusion Model (EDM) (Hoogeboom et al., 2022). We denote a 3D molecule with $M$ atoms as $\boldsymbol{z} = (\boldsymbol{x}, \boldsymbol{h})$, where $\boldsymbol{x} = (\boldsymbol{x}_1, \cdots, \boldsymbol{x}_M) \in \mathbb{R}^{3M}$ represents the atomic positions, and $\boldsymbol{h} = (\boldsymbol{h}_1, \cdots, \boldsymbol{h}_M) \in \{\boldsymbol{e}_0, \cdots, \boldsymbol{e}_H\}^M$ encodes the atom type information. Each $\boldsymbol{h}_i$ is a one-hot vector of dimension $H$, corresponding to the type of the atom $i$, where $H$ is the total number of distinct atom types in the dataset.

### 2.1.1 TWO-PHASE GENERATIVE PROCESS

In contrast to traditional diffusion based model, UniGEM adopts diffusion process to generate only the atomic coordinates $\boldsymbol{x}$ rather than the whole molecule $\boldsymbol{z}$. We define the forward process for adding noise to the molecular coordinates as follows:

$$q(\boldsymbol{x}_{0:T}) = q(\boldsymbol{x}_0) \prod_{t=1}^{t_n} q(\boldsymbol{x}_t|\boldsymbol{x}_{t-1}) \prod_{t=t_n+1}^{T} q(\boldsymbol{x}_t|\boldsymbol{x}_{t-1}), \tag{1}$$

where $t_n$ denotes the nucleation time to indicate when the molecule forms its scaffold. We define the growth phase as $t \in [0, t_n]$ and the nucleation phase as $t \in [t_n, T]$. During the growth phase, we will incorporate atom type and property information, as these attributes are clearly defined at this stage. The conditional distribution is defined as $q(\boldsymbol{x}_t|\boldsymbol{x}_0) = \mathcal{N}_x(\boldsymbol{x}_t|\alpha_t \boldsymbol{x}_0, \sigma_t^2 \boldsymbol{I})$, $t = 1, \cdots, T$, where $\alpha_t$ and $\sigma_t$ represent the noise schedule following Hoogeboom et al. (2022) and satisfy the condition $\alpha_t^2 + \sigma_t^2 = 1$, with $\alpha_t$ decreases monotonically from 1 to 0. $\boldsymbol{z}_0 = (\boldsymbol{x}_0, \boldsymbol{h}_0)$ represent an unperturbed molecule in the dataset. $\mathcal{N}_x$ represents the Gaussian distribution in the zero center-of-mass (CoM) subspace satisfying $\sum_i^M \boldsymbol{x}_i = \boldsymbol{0}$, to ensure translation equivariance.

In UniGEM, we train a unified model, denoted as $\phi_\theta$, that is capable of both generative and predictive tasks. An overview is provided in Figure 1. Following previous molecular generative mod-

els (Hoogeboom et al., 2022; Wu et al., 2022; Xu et al., 2023), we adopt an EGNN architecture (Satorras et al., 2021) for our implementation.

For the coordinate generation task, we train the model to predict noise from the noisy inputs using a mean squared error (MSE) loss at each timestep:

$$\mathcal{L}_t^{(x)} = \mathbb{E}_{q(\boldsymbol{x}_0, \boldsymbol{x}_n)} \|\boldsymbol{\phi}_\theta^{(x)}(\boldsymbol{x}_t, t) - \boldsymbol{\epsilon}_t\|^2, \quad \boldsymbol{\epsilon}_t = (\boldsymbol{x}_t - \alpha_t \boldsymbol{x}_0)/\sigma_t, \quad t \in [1, T] \tag{2}$$

where $\boldsymbol{\epsilon}_t$ is the standard Gaussian noise injected to $\boldsymbol{x}_0$ and $\boldsymbol{\phi}_\theta^{(x)}(\boldsymbol{x}_t, t)$ is the equivariant output of the backbone network. The denoising loss is equivalent to the score estimation loss $\simeq \mathbb{E}_{q(\boldsymbol{x}_0, \boldsymbol{x}_n)} \|\boldsymbol{\phi}_\theta^{(x)}(\boldsymbol{x}_t, t) - (-\sigma_t \nabla_{\boldsymbol{x}_t} \log q(\boldsymbol{x}_t))\|^2$, where $\nabla_{\boldsymbol{x}_t} \log q(\boldsymbol{x}_t))$ refers to the score function. This is derived through the equivalence between score matching and conditional score matching (Vincent, 2011b), with a formal proof presented in Ni et al. (2024); Feng et al. (2023a). The learned score function is crucial for reversing the noise addition process for coordinate generation.

For the atom type prediction task, we use an $L_1$ loss for regressing the one-hot vector of the atom types:

$$\mathcal{L}_t^{(h)} = \mathbb{E}_{q(\boldsymbol{x}_0, \boldsymbol{x}_n)} |\boldsymbol{\phi}_\theta^{(h)}(\boldsymbol{x}_t, t) - \boldsymbol{h}|, \quad t \in [1, t_n], \tag{3}$$

where $\boldsymbol{h}$ is the one-hot vector of atom types corresponding to the coordinates $\boldsymbol{x}_0$ in the dataset, and $\boldsymbol{\phi}_\theta^{(h)}$ is the invariant output from the backbone network with an additional two-layer MLP prediction head.

In the property prediction task, we denote the property of the molecule $\boldsymbol{z}_0$ as $c$, and utilize an $L_2$ loss for regressing the property:

$$\mathcal{L}_t^{(c)} = \mathbb{E}_{q(\boldsymbol{x}_0, \boldsymbol{x}_n)} \|\boldsymbol{\phi}_\theta^{(c)}(\boldsymbol{x}_t, t) - c\|^2, \quad t \in [1, t_n], \tag{4}$$

where $\boldsymbol{\phi}_\theta^{(c)}(\boldsymbol{x}_t, t)$ is the output from the backbone network with another two-layer MLP prediction head. For $t \in [t_n, T]$, we set $\mathcal{L}_t^{(c)}$ and $\mathcal{L}_t^{(h)}$ as zero.

In this way, we can seamlessly integrate predictive and generative tasks using a single model, and avoid inconsistencies of the two tasks when the molecule structure is not fully formed. However, employing traditional diffusion strategies to train this unified model would fail. To overcome this challenge, we have developed a series of training strategies aimed at optimizing the balance between the two phases and maximizing the benefits of integration.

### 2.1.2 TRAINING STRATEGIES

In our framework, the growth phase occupies a small portion of the overall training process, with the best-performing configuration being $t_n/T = 0.01$. If we follow the standard diffusion training procedure and sample time steps uniformly, the number of iterations for the predictive task only takes 1% of the total training process , which will significantly degrade the model's performance on this task. Therefore, to ensure sufficient training for the predictive task, we oversample the time steps in the growth phase. Thus, the final loss can be written as the following equation:

$$\mathcal{L} = \mathbb{E}_{t \sim \frac{1}{2} U(1, t_n] + \frac{1}{2} U(t_n, T]} \left( \mathcal{L}_t^{(x)} + \mathcal{L}_t^{(h)} + \mathcal{L}_t^{(c)} \right) \tag{5}$$

However, we observed that oversampling can lead to imbalanced training across the full range of time steps, which adversely affects the quality of the generative process. To address this issue, we propose a multi-branch network architecture. The network shares parameters in the shallow layers but splits into two branches in the deeper layers, with separate sets of parameters for each branch. These branches are activated in different phases of training: one branch is dedicated to the nucleation phase, while the other handles the growth phase, as shown in Figure 3. This design ensures that the predictive and generative tasks are trained effectively without negatively influencing each other.

### 2.1.3 INFERENCE OF UNIGEM

In UniGEM, molecular generation is achieved by reversing the forward diffusion process through a Markov chain to reconstruct atomic coordinates, followed by predicting atom types based on the generated coordinates. Specifically, we express the reverse process as $p_\theta(\boldsymbol{x}_{0:T}) =$

$p_\theta(\boldsymbol{x}_T) \prod_{t=1}^{T} p_\theta(\boldsymbol{x}_{t-1}|\boldsymbol{x}_t)$, where the noise at each step is approximated by the trained neural network $\phi_\theta^{(x)}(\boldsymbol{x}_t, t)$. The reverse transition at each time step is given by:

$$p_\theta(\boldsymbol{x}_{t-1}|\boldsymbol{x}_t) = \mathcal{N}_x\left(\boldsymbol{x}_{t-1} \middle| \frac{1}{\alpha_{t|t-1}}\boldsymbol{x}_t - \frac{\sigma_{t|t-1}^2}{\alpha_{t|t-1}\sigma_t}\phi_\theta^{(x)}(\boldsymbol{x}_t, t), (\frac{\sigma_{t|t-1}\sigma_{t-1}}{\sigma_t})^2\boldsymbol{I}\right). \tag{6}$$

Here $\alpha_{t|t-1} = \alpha_t/\alpha_{t-1}$ and $\sigma_{t|t-1}^2 = \sigma_t^2 - \alpha_{t|t-1}^2\sigma_{t-1}^2$. The prior distribution is approximated by $p(\boldsymbol{x}_N) = \mathcal{N}_x(x_N|\boldsymbol{0}, \boldsymbol{I})$.

For the molecular generation task, the atom types are predicted at the final time step, producing a complete molecule as $\boldsymbol{z} = (\boldsymbol{x}_0, \phi_\theta^{(h)}(\boldsymbol{x}_0, 0))$, as shown in Figure 4. For property prediction, the inference is done with the time step fixed at zero during testing, i.e., using the output $\phi_\theta^{(c)}(\boldsymbol{x}_0, 0)$. Notably, this approach incurs no additional computational overhead for either the generative or predictive tasks. The total inference time is the same as the baseline, ensuring that both tasks are performed efficiently without increasing the overall runtime.

## 2.2 THEORETICAL ANALYSIS

In this section, we present our theoretical results addressing two questions: first, why diffusion is capable of learning representations while the direct multi-task approach fails; and second, why the proposed UniGEM effectively enhances performance in the challenging generation task.

### 2.2.1 INCONSISTENCY BETWEEN MOLECULE GENERATION AND PROPERTY PREDICTION

Our study uses an Information Maximization (InfoMax) perspective (Linsker, 1988; Oord et al., 2018), which serves as a criterion of the representation quality. According to the InfoMax theory, effective latent representations $\zeta_t$ in the diffusion-based molecule generation approach are achieved by maximizing the mutual information (MI) between the original molecular coordinates $\boldsymbol{x}_0$ and $\zeta_t$. $\zeta_t$ are derived from the noisy coordinates $\boldsymbol{x}_t$ in the intermediate layers of the denoising network.

**Theorem 2.1.** *The mutual information between $\boldsymbol{x}_0$ and $\zeta_t$ can be expressed as follows, with a subsequent lower bound:*

$$\begin{aligned} I(\zeta_t, \boldsymbol{x}_0) &= I(\boldsymbol{x}_0; \boldsymbol{x}_t) - \mathbb{E}_{q(\boldsymbol{x}_t, \zeta_t)}\left[D_{KL}(q(\boldsymbol{x}_0|\boldsymbol{x}_t)\|q(\boldsymbol{x}_0|\zeta_t))\right] \\ &\geq I(\boldsymbol{x}_0; \boldsymbol{x}_t) - \mathbb{E}_{q(\boldsymbol{x}_t, \zeta_t)}\left[D_{KL}(q(\boldsymbol{x}_0|\boldsymbol{x}_t)\|p(\boldsymbol{x}_0|\zeta_t))\right], \end{aligned} \tag{7}$$

*where $q(x_0, x_t)$ are data distribution defined by the forward process of diffusion, $q(\zeta_t|\boldsymbol{x}_t) = \delta_{g_\theta(\boldsymbol{x}_t)}$ and $p(\boldsymbol{x}_0|\zeta_t)$ represent the estimated representation and denoising distributions by the denoising network. In practice, our denoising network models $g_\theta(\boldsymbol{x}_t)$ and the mean of the denoising distribution $\mathbb{E}_p(\mathbf{x}_0|g_\theta(\mathbf{x}_t)) := \frac{\mathbf{x}_t - \sigma_t\varphi_\theta(\mathbf{x}_t, t)}{\alpha_t}$.*

As we show in appendix F.3, when $p(\boldsymbol{x}_0|\zeta_t)$ and $q(\boldsymbol{x}_0|\boldsymbol{x}_t)$ follow Gaussian distributions with the same variance $\sigma$, minimizing the KL divergence is equivalent to minimizing the denoising diffusion loss. Thus, the second term $\mathbb{E}_{q(\boldsymbol{x}_t, \zeta_t)}\left[D_{\mathrm{KL}}(q(\boldsymbol{x}_0|\boldsymbol{x}_t)\|p(\boldsymbol{x}_0|\zeta_t))\right]$ can be minimized during the diffusion training process. This suggests that effective representations can be implicitly learned throughout the diffusion training.

However, according to the data processing inequality (Thomas & Cover, 2006), the first term $I(\boldsymbol{x}_0; \boldsymbol{x}_t)$ decreases monotonically as $t$ increases, eventually approaching zero. As a result, $I(\zeta_t, \boldsymbol{x}_0)$ becomes small for large values of $t$, indicating that it is increasingly challenging to learn meaningful representations from noisy molecules. This explains the failure of multi-task learning in such cases. In contrast, when $t$ is small, $I(\zeta_t, \boldsymbol{x}_0)$ is larger, suggesting that the two tasks can align to facilitate learning robust representations, which emphasizes the need for a two-phase modeling approach.

### 2.2.2 THEORETICAL ANALYSIS ON GENERATION ERROR

To further investigate why UniGEM improves performance on the generation task, we conduct a theoretical analysis of the generation error. Specifically, we derive the upper bound of the generative error for two molecular generation approaches: UniGEM and traditional diffusion-based models, based on results from Chen et al. (2023b). Complementary details are provided in appendix G.

**Theorem 2.2** (Generative Error Analysis). *With mild assumptions on the data distribution $q$ provided in appendix G.1, the total variation error between the UniGEM generated data distribution $p_\theta(\boldsymbol{z}_0)$ and ground-truth data distribution $q(\boldsymbol{z}_0)$ is bounded by the following terms.*

$$TV(p_\theta(\boldsymbol{z}_0), q(\boldsymbol{z}_0)) \lesssim \underbrace{\sqrt{KL(q(\boldsymbol{x}_0)||p_\theta(\boldsymbol{x}_T))}e^{-\tilde{T}}}_{\text{prior distribution error}} + \underbrace{(L_x\sqrt{d_x l} + L_x m_x l)\sqrt{\tilde{T}}}_{\text{discretization error}} + \underbrace{\sqrt{l\sum_{t=1}^{T}\mathcal{L}_t^{(x)}}}_{\substack{\text{coordinate score} \\ \text{estimation error}}} + \underbrace{\frac{1}{2}\mathbb{E}_{q(\boldsymbol{x}_0)}\mathcal{L}^{(h)}(\boldsymbol{x}_0)}_{\substack{\text{atom type} \\ \text{estimation error}}}. \quad (8)$$

*For traditional diffusion models generating coordinates and atom types simultaneously with distribution $r_\theta(\boldsymbol{z}_0)$, the error bound becomes:*

$$TV(r_\theta(\boldsymbol{z}_0), q(\boldsymbol{z}_0)) \lesssim \underbrace{\sqrt{KL(q(\boldsymbol{z}_0)||r_\theta(\boldsymbol{z}_T))}e^{-\tilde{T}}}_{\text{prior distribution error}} + \underbrace{(L_z\sqrt{d_z l} + L_z m_z l)\sqrt{\tilde{T}}}_{\text{discretization error}} + \underbrace{\sqrt{l\sum_{t=1}^{T}\mathcal{L}_t^{(x|h)}}}_{\substack{\text{coordinate score} \\ \text{estimation error}}} + \underbrace{\sqrt{l\sum_{t=1}^{T}\mathcal{L}_t^{(h|x)}}}_{\substack{\text{atom type score} \\ \text{estimation error}}}. \quad (9)$$

*Here for $y \in \{x, z\}$, $m_y^2$ is the second moment of $q(y_0)$; $L_y$ is the Lipschitz constant of $\nabla \ln q(y_t)$, $\forall t \geq 0$. $\tilde{T}$ is the terminal timestep; $T$ is the discretization steps. We assumed the step size $l := \tilde{T}/T \lesssim 1/L_y$ and $L_y \geq 1$.*

Our theoretical analysis identifies four factors contributing to generation errors in both UniGEM and traditional diffusion-based models. The prior distribution and discretization errors both decrease with lower data dimensionality. Since UniGEM only generates atom coordinates, unlike traditional models that jointly generate both coordinates and atom types, UniGEM deals with significantly smaller data dimensionality, leading to reduced errors. As for the latter two errors, it's harder to compare through theory alone, so we supplement with experimental results on QM9 datasets in Table 1, showing both of them are smaller in UniGEM.

The reduced coordinate and atom type errors in UniGEM can be intuitively explained. A critical issue with traditional diffusion-based models is their treatment of discrete atom types as continuous variables, which can lead to instability during generation. For example, in our experiments, we observe that even at the later stage of generation, the predicted atom type (determined by the highest probability) oscillates between two categories, indicating a suboptimal learned distribution. Since the coordinates in the next step depend on both the current coordinate and atom type, this oscillation introduces additional errors in coordinate estimation. In contrast, UniGEM avoids this issue, as it only generates the coordinates, potentially leading to more accurate and stable coordinate predictions.

Table 1: Comparisons of coordinate and atom type errors.

|  | UniGEM | EDM |
|---|---|---|
| Atom type | **1.3E-5**(0.0001) | 0.0077(0.0012) |
| Coordinate | **0.2247**(0.0408) | 0.2430(0.0434) |

For the atom type error, UniGEM approaches this as a prediction task, which is significantly simpler than the generation approach used in traditional methods, especially after the nucleation phase when molecule scaffolds have formed. Therefore, it is not surprising that the atom type error in UniGEM is smaller.

Regarding the above points, we can conclude that it is both reasonable and necessary to decouple the discrete atom types from the joint generation process and handle them through a separate prediction task.

## 3 EXPERIMENTS

### 3.1 MAIN EXPERIMENTS

**Datasets** In our experiments, we utilize two widely used datasets. The QM9 dataset (Ruddigkeit et al., 2012; Ramakrishnan et al., 2014) serves both molecular generation and property prediction tasks, comprising approximately 134,000 small organic molecules with up to nine heavy atoms, each annotated with quantum chemical properties such as LUMO (Lowest Unoccupied Molecular Orbital), HOMO (Highest Occupied Molecular Orbital), HOMO-LUMO gap, polarizability ($\alpha$). These properties are used as ground-truth labels for property prediction. In contrast, the GEOM-

Table 2: Comparison of generation performance on metrics, including atom stability, molecule stability, validity, and validity*uniqueness. Higher values indicate better performance.

| #Metrics | QM9 | | | | GEOM-Drugs | |
|---|---|---|---|---|---|---|
| | Atom sta(%) | Mol sta(%) | Valid(%) | V*U(%) | Atom sta(%) | Valid(%) |
| E-NF | 85.0 | 4.9 | 40.2 | 39.4 | - | - |
| G-Schnet | 95.7 | 68.1 | 85.5 | 80.3 | - | - |
| EDM | 98.7 | 82.0 | 91.9 | 90.7 | 81.3 | 92.6 |
| GDM | 97.6 | 71.6 | 90.4 | 89.5 | 77.7 | 91.8 |
| EDM-Bridge | 98.8 | 84.6 | 92.0 | 90.7 | 82.4 | 92.8 |
| GeoLDM | 98.9 | 89.4 | 93.8 | 92.7 | 84.4 | **99.3** |
| UniGEM | **99.0** +0.3% | **89.8** +7.8% | **95.0** +3.1% | **93.2** +2.5% | **85.1** +3.8% | 98.4 +5.8% |

Drugs dataset (Axelrod & Gomez-Bombarelli, 2022) focuses on drug-like molecules, providing a large-scale collection of molecular conformers. It includes around 430,000 molecules, with sizes ranging up to 181 atoms and an average of 44.4 atoms per molecule.

The splitting strategies for both benchmarks follow the previous practices. For the QM9 dataset, we adopt the same split as in prior methods (Hoogeboom et al., 2022; Satorras et al., 2021; Anderson et al., 2019), dividing the data into training (100K), validation (18K), and test (13K) sets. Similarly, for the GEOM-Drugs dataset, we follow the approach outlined in Hoogeboom et al. (2022) to randomly split the dataset into training, validation, and test sets in an 8:1:1 ratio.

**Implementation Details** We adopt EGNN (Satorras et al., 2021) as our backbone and modify it into a multi-branch network. As illustrated in Figure 3, different branches handle the diffusion loss at different ranges of time steps, with shared layers preceding the branches. In the branch for $t \leq t_n$, we add predictive losses, including molecular property prediction and atom type prediction. During training, we apply uneven sampling for each batch. For the first half of the batch, the sampling range for $t$ is $[t_n, T]$, and for the second half, the sampling range for $t$ is $[0, t_n]$, ensuring that each branch receives the same samples and is sufficiently trained. All losses are optimized in parallel, with each loss weighted equally at 1. We implement UniGEM with 9 layers, each consisting of 256 dimensions for the hidden layers. The shared layer count is set to 1, while each separate branch, corresponding to different ranges of time steps, has 8 layers. Optimization is performed using the Adam optimizer, with a batch size of 64 and a learning rate of $10^{-4}$. We train UniGEM on QM9 for 2,000 epochs, while training on the GEOM-Drugs dataset is limited to 13 epochs due to the larger scale of its training data. We set the nucleation time $t_n$ to 10 and the total time steps $T$ to 1000, ensuring that the prediction of atom types and property occurs within the first 10 time steps. The atom type and molecular property predicted in the last time step are used as the final predictive result.

### 3.1.1 EXPERIMENTAL RESULTS FOR MOLECULE GENERATION

**Baselines** UniGEM is implemented based on the codebase of a classic 3D molecular diffusion algorithm, EDM (Hoogeboom et al., 2022). Thus, our baseline comprises EDM and its variants, GDM (Hoogeboom et al., 2022), EDM-Bridge (Wu et al., 2022) and GeoLDM (Xu et al., 2023). GDM utilizes non-equivariant networks for training, EDM-Bridge enhances EDM through the technique of diffusion bridges. GeoLDM introduces an additional autoencoder to encode molecular structures into latent embeddings, where the diffusion process is conducted in the latent space. Additionally, we include G-Schnet (Gebauer et al., 2019), an autoregressive generation method, and Equivariant Normalizing Flows (E-NF) (Garcia Satorras et al., 2021) in our comparisons.

**Metrics** Following the approach of these baselines, we sample 10,000 molecules and evaluate atom stability, molecule stability, validity, and uniqueness of valid samples. Specifically, we utilize the distances between each pair of atoms to predict the bond type-whether it is single, double, triple, or non-existent. Atom stability is calculated as the ratio of atoms exhibiting correct valency, while molecule stability reflects the fraction of generated molecules in which each atom maintains stability. Validity and uniqueness are assessed using RDKit by converting the 3D molecular structures into SMILES format, with uniqueness determined by calculating the ratio of unique generated molecules among all valid samples after removing duplicates.

Table 3: Performance (MAE, ↓) on 6 property prediction tasks in QM9. The best results are in bold.

| Task (Units) | $\alpha$ (bohr$^3$) | $\Delta\epsilon$ (meV) | $\epsilon_{\text{HOMO}}$ (meV) | $\epsilon_{\text{LUMO}}$ (meV) | $\mu$ (D) | $C_v$ ($\frac{cal}{mol \cdot K}$) |
|---|---|---|---|---|---|---|
| EGNN | 0.071 | 48 | 29 | 25 | 0.029 | 0.031 |
| *GraphMVP* | 0.070 | 46.9 | 28.5 | 26.3 | 0.031 | 0.033 |
| *3D Infomax* | 0.075 | 48.8 | 29.8 | 25.7 | 0.034 | 0.033 |
| *GEM* | 0.081 | 52.1 | 33.8 | 27.7 | 0.034 | 0.035 |
| *3D-EMGP* | **0.057** | 37.1 | 21.3 | 18.2 | 0.020 | 0.026 |
| UniGEM | 0.060 -15.5% | **34.5** -28.1% | **20.9** -27.9% | **16.7** -33.2% | **0.019** -34.5% | **0.023** -25.8% |

**Results** The results in Table 2 show that UniGEM consistently outperforms all baselines across nearly all evaluation metrics for both QM9 and GEOM-Drugs, with the gray cells indicating the molecule generator used in our approach. Notably, compared to EDM variants, UniGEM is significantly simpler, as it neither relies on prior knowledge nor requires additional training for an autoencoder, yet it achieves superior performance compared to EDM-Bridge and GeoLDM, highlighting UniGEM's superiority. A visualization of the generation process is presented in Figure 5. To demonstrate the flexibility of UniGEM in adapting to various generation algorithms, we apply UniGEM to Bayesian Flow Networks (BFN) Graves et al. (2023); Song et al. (2024a) and achieve SOTA results on the QM9 dataset, as detailed in Appendix B.1.

Furthermore, we test UniGEM on conditional generation task in Appendix B.2, by using pre-trained property prediction module as guidance during sampling. This approach eliminates the need for retraining a conditional generation model and avoids additional guidance networks.

### 3.1.2 EXPERIMENTAL RESULTS FOR MOLECULAR PROPERTY PREDICTION

**Baselines** The property prediction module of UniGEM is built on the widely used equivariant neural network EGNN (Satorras et al., 2021). We compare UniGEM against the EGNN model trained from scratch, referred to as EGNN without confusion, as well as several recent pre-training methods, that have demonstrated superiority in learning molecular representations, thereby improving the property prediction performance. These methods include GraphMVP (Liu et al., 2021), 3D Infomax (Stärk et al., 2022), GEM (Fang et al., 2022), and 3D-EMGP (Jiao et al., 2023). It is important to note that all baseline methods utilize the same backbone and data splits to ensure a fair comparison.

**Results** We evaluated UniGEM on six properties of the QM9 dataset, using the Mean Absolute Error (MAE) on the test set as the evaluation metric. As shown in Table 3, UniGEM significantly outperforms EGNN trained from scratch, with the results highlighted in the grey cell, demonstrating the effectiveness of unified modeling. Surprisingly, UniGEM also surpasses most advanced pre-training methods (*italicized*), even though they utilize additional large-scale pre-training datasets. This highlights the strength of its unified approach, which effectively leverages the capabilities of molecular representation learning during the generation process, without requiring additional data and pre-training steps. The relationship with denoising pre-training is discussed in Appendix B.3.

### 3.1.3 COMPARISONS WITH GENERAL UNIFIED APPROACHES

In this section, we compare UniGEM with two general approaches for unifying generation and property prediction tasks. Table 4 outlines these approaches: the first treats generation and property prediction as a multi-task joint training network, referred to as 'Multi-task'. Additionally, we incorporate the property prediction task into a pre-trained generation network for continued training, specifically freezing the backbone to preserve the network's generation capability, referred to as 'Gen Pre-train'.

**Results** presented in Table 4 demonstrate that both simple approaches fail to achieve better results than the baseline. The performance of the multi-task approach aligns with our previous analysis: there is an inconsistency between the generation process and the property prediction task, and simple integration negatively impacts the performance of both tasks. While 'Gen Pre-train' preserves the model's generation capabilities by freezing the backbone, it also restricts the network's expressiveness, which in turn limits its performance in property prediction.

Table 4: Comparison of UniGEM with other general unified approaches on both generation and property prediction tasks. The best results are in bold.

| | Generation | | | | Prop Pred |
|---|---|---|---|---|---|
| #Metrics | Atom sta(%) | Mol sta(%) | Valid(%) | V*U(%) | $\epsilon_{LUMO}$ |
| EDM | 98.7 | 82.0 | 91.9 | 90.7 | - |
| EGNN | - | - | - | - | 25 |
| Multi-task | 98.0 | 76.0 | 89.9 | 88.8 | 51.8 |
| Gen Pre-train | 98.7 | 82.0 | 91.9 | 90.7 | 51.2 |
| UniGEM | **99.0** | **89.8** | **95.0** | **93.2** | **16.7** |

Table 5: Ablation study about the impact of different predictive loss functions on the generation and prediction performance.

| | Generation | | | | Prop Pred |
|---|---|---|---|---|---|
| #Metrics | Atom sta(%) | Mol sta(%) | Valid(%) | V*U(%) | $\epsilon_{LUMO}$ |
| EDM | 98.7 | 82.0 | 91.9 | 90.7 | - |
| EGNN | - | - | - | - | 25 |
| MPP | 98.7 | 85.7 | 94.2 | 92.6 | 20.6 |
| ATP | 98.7 | 89.3 | 94.6 | 92.8 | - |
| MPP+ATP | **99.0** | **89.8** | **95.0** | **93.2** | **16.7** |

## 3.2 ABLATION STUDIES

We conduct three ablation studies to assess the impact of predictive loss, time sampling strategy, and nucleation time setting using the QM9 dataset. These studies include validation on both generation and property prediction tasks, comparing our results to the generation baseline EDM and the property prediction baseline EGNN trained from scratch.

### 3.2.1 IMPACT OF PREDICTIVE LOSS ON PERFORMANCE

In UniGEM, we incorporate two types of predictive loss: atom type prediction loss and molecular property prediction loss. To validate their impact, we compare our model against two baselines: one utilizing only atom type prediction loss (referred to as ATP) and the other using only molecular property prediction loss (referred to as MPP). The comparison results are presented in Table 5, with LUMO selected as the target property for performance evaluation.

**Results** are shown in Table 5, both ATP and MPP enhance generation performance compared to previous baseline models, with ATP demonstrating a greater improvement than MPP. The combination of these two losses functions yields the best results, as evidenced by UniGEM. These findings underscore the importance of both predictive losses, particularly highlighting the critical role of atom type prediction.

In molecular property prediction, MPP enhances the original performance, and incorporating the atom type prediction loss further improves results. This improvement may stem from atom type prediction facilitating the learning of better molecular representation, which ultimately benefits property prediction.

### 3.2.2 ANALYSIS OF TRAINING STRATEGIES IN UNIGEM

To ensure sufficient training steps for property prediction, we oversample the time steps after the nucleation time. Additionally, we design a multi-branch network to mitigate the negative impact of oversampling on generation performance. We conduct ablation studies to demonstrate the necessity of oversampling and branch splitting strategies in ensuring superior performance for both generation and prediction tasks.

**Results** are shown in Table 6, where the first two rows represent the baselines, and the third row indicates that single-branch with normal time sampling(referred to as 'Normal Sampling (S)') achieves higher generation performance than the baseline but exhibits degraded prediction performance. This suggests that the proportion of structured time is too small for sufficient property prediction training.

Table 6: Performance of different training strategies on the generation and prediction tasks, with the best results highlighted in bold.

| #Metrics | Generation | | | | Prop Pred |
|---|---|---|---|---|---|
| | Atom sta(%) | Mol sta(%) | Valid(%) | V*U(%) | $\epsilon_{LUMO}$ |
| EDM | 98.7 | 82.0 | 91.9 | 90.7 | - |
| EGNN | - | - | - | - | 25 |
| Normal Sampling(S) | 98.9 | 89.7 | 94.8 | 92.9 | 40.5 |
| Oversampling(S) | 98.3 | 80.1 | 90.1 | 88.8 | 19.2 |
| UniGEM | **99.0** | **89.8** | **95.0** | **93.2** | **16.7** |

In the fourth row, adding oversampling to the molecular growth phase(referred to as 'Oversampling (S)') improves property performance compared to the baseline, but lowers generation performance due to the effects of unbalanced time step sampling. Finally, in UniGEM (last row), we change the backbone to a multi-branch architecture, allowing different branches to handle different time steps. This adjustment eliminates the negative impact of time step oversampling, resulting in superior performance on both tasks.

### 3.2.3 Effect of Nucleation Time

Nucleation time is defined as the moment when the molecular scaffold forms, after which the molecular coordinates experience only minor adjustments. In practical applications, accurately determining the true nucleation time is challenging. Therefore, we conduct an empirical analysis to assess its impact on model performance. We set the total training step to T=1000, and compare models with nucleation times of 1, 10, and 100.

**Results** are outlined in Figure 2. Across all generation and property prediction criteria, a nucleation time of 10 achieves optimal performance. Setting the nucleation time too large may incorporate time steps prior to the complete formation of the molecular scaffold, leading to suboptimal outcomes. Conversely, using a nucleation time that is too small also degrades performance, potentially due to insufficient input noise, which is crucial for learning a more robust type and property mapping. From the perspective of force learning interpretation of the denoising task, as discussed in Appendix F.4, a moderate level of noise is advantageous for capturing molecular representations, thus enhancing the predictive tasks.

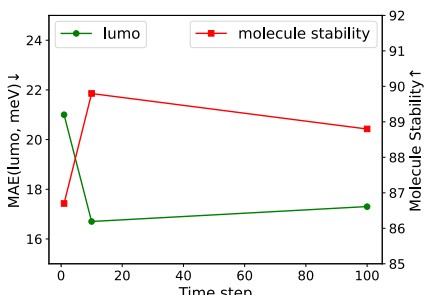

Figure 2: Comparison of performance across different nucleation time selections for both the generation and property prediction tasks.

## 4 Conclusion

This paper introduces UniGEM, the first effective unified model that significantly improves the performance of both molecular generation and property prediction tasks. The underlying philosophy is that these traditionally separate tasks are highly correlated due to their reliance on effective molecular representations. The traditional inconsistencies of these tasks are overcome by a two-phase generative process with atom type and property prediction losses activated in the growth phase. UniGEM's enhanced performance is supported by solid theoretical analysis and comprehensive experimental studies. We believe that the innovative two-phase generation process and its corresponding models offer a new paradigm that may inspire the development of more advanced molecule generation frameworks and benefit more specific applications of molecular generation.

### Acknowledgements

This work is supported by Beijing Academy of Artificial Intelligence (BAAI).

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

# Appendix

## Table of Contents

# A   COMPLIMENTARY FIGURES

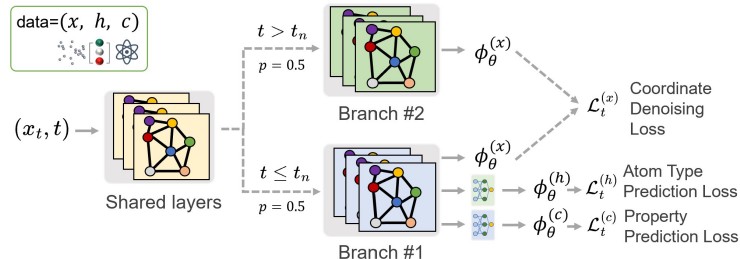

Figure 3: The training process of UniGEM.

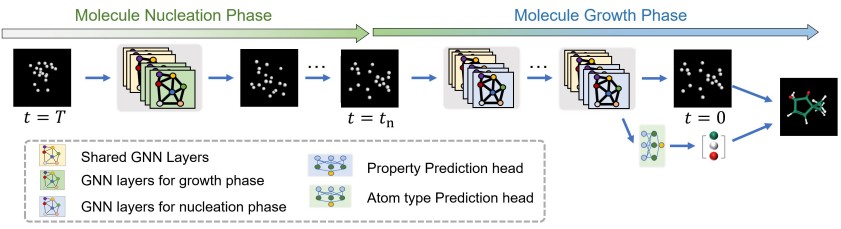

Figure 4: The molecular generative process of UniGEM.

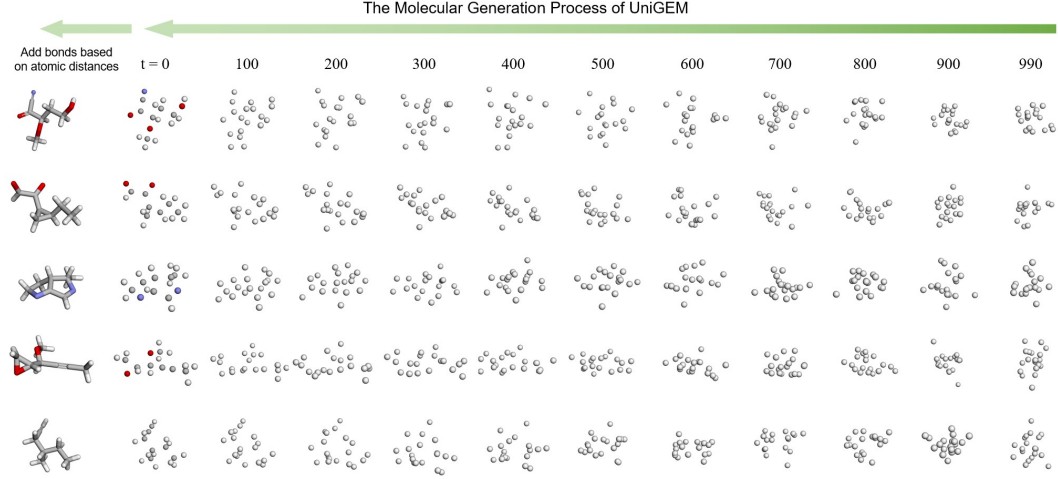

Figure 5: A visualization of the generation process of UniGEM.

# B   SUPPLEMENTARY EXPERIMENTS

## B.1   APPLYING UNIGEM TO ADVANCED GENERATIVE METHODS

As UniGEM is a flexible framework that can adapt to various generation algorithms, we expect it to produce better results when adapted to more advanced generation algorithms. To illustrate this, we conducted additional experiments with UniGEM using the current SOTA generation algorithm, BFN (Graves et al., 2023; Song et al., 2024a). Specifically, we replaced EDM's coordinate generation schedule with BFN's coordinate generation algorithm. The results, presented in the table 7, highlight two key findings:

Table 7: Molecular Generation Performance on QM9.

| | Mol sta(%) | Atom sta(%) | Valid(%) | V*U(%) |
|---|---|---|---|---|
| E-NF | 4.9 | 85.0 | 40.2 | 39.4 |
| G-Schnet | 68.1 | 95.7 | 85.5 | 80.3 |
| GDM | 71.6 | 97.6 | 90.4 | 89.5 |
| EDM-Bridge | 84.6 | 98.8 | 92.0 | 90.7 |
| EquiFM | 88.3 | 98.9 | 94.7 | 93.5 |
| GFMDiff(Xu et al., 2024) | 87.7 | 98.9 | 96.3 | **95.1** |
| GeoLDM | 89.4 | 98.9 | 93.8 | 92.7 |
| EDM | 82.0 | 98.7 | 91.9 | 90.7 |
| UniGEM (w/ EDM) | 89.8 +7.8% | 99.0 +0.3% | 95.0 +3.1% | 93.2 +2.5% |
| GeoBFN | 90.9 | 99.1 | 95.3 | 93.0 |
| UniGEM(w/ BFN) | **93.7** +2.8% | **99.3** +0.2% | **97.3** +2.0% | 93.0 +0.0% |

Firstly, UniGEM can still achieve a significant improvement on top of a stronger generation algorithm, showcasing its robustness and adaptability across diverse frameworks.

Secondly, by leveraging diffusion for continuous coordinate generation and classification for discrete atom types, UniGEM effectively addresses the simultaneous generation of discrete and continuous variables. Its decoupled and straightforward approach outperforms GeoBFN (Song et al., 2024a), which specifically tailors its generative algorithm for discrete atom types and discretized atom charge variables.

## B.2 CONDITIONAL GENERATION

To assess the effectiveness of the proposed method, we evaluate it on a property-conditioned generation task using the QM9 dataset. This task provides a more rigorous test of the model's ability to learn representations related to specific conditions while also capturing the underlying data distribution—objectives that align closely with the goals of the proposed method.

Unlike traditional methods, UniGEM integrates property prediction directly into its framework, enabling the use of its pre-trained property prediction module to guide molecular coordinate updates. This approach **bypasses the need to retrain a conditional generation model and eliminates the requirement for an additional guidance network**, as both property prediction and generation share the same network.

The update step for the molecular coordinates is given by $x_{t-1} = \frac{1}{\alpha_{t|t-1}}x_t - \frac{\sigma^2_{t|t-1}}{\alpha_{t|t-1}\sigma_t}\phi_\theta^{(x)}(x_t, t) - \lambda\nabla L_t^{(c)} + \frac{\sigma_{t|t-1}\sigma_{t-1}}{\sigma_t}\epsilon$, where $L_t^{(c)}$ is the property prediction loss of UniGEM.

Table 8: Performance of conditional generation on QM9 properties.

| | $\epsilon_{\text{LUMO}}$ (eV) | $\epsilon_{\text{HOMO}}$ (eV) | $\Delta\epsilon$ (eV) | $\mu$ (D) | $\alpha$ (bohr$^3$) | $C_v$ ($\frac{cal}{mol \cdot K}$) |
|---|---|---|---|---|---|---|
| Conditional EDM(Average MAE↓) | 0.606 | 0.356 | 0.665 | 1.111 | 2.76 | 1.101 |
| Guided UniGEM (Average MAE↓) | **0.592** | **0.233** | **0.511** | **0.805** | **2.22** | **0.873** |
| Guided UniGEM (Mol sta(% ↑)) | 81.5 | 86.3 | 80.4 | 83.2 | 83.0 | 80.9 |
| Guided UniGEM (stable MAE↓) | 0.596 | 0.285 | 0.499 | 0.764 | 2.15 | 0.877 |

As shown in Table 8, UniGEM achieves performance better than that of the conditional EDM trained explicitly with property conditions as input. This demonstrates the flexibility and capability of UniGEM in property-conditioned molecular generation.

To improve the evaluation of conditional property prediction tasks, we introduce two new metrics: Mol Stable and Stable MAE, addressing limitations in traditional evaluation methods. Existing

approaches often rely on an auxiliary property classifier (e.g., EGNN) to generate pseudo-labels for the generated molecules, comparing these labels with the target conditions to assess generation quality. However, these classifiers only utilize atom types and coordinates, meaning that even invalid molecules can yield seemingly accurate property predictions. To simultaneously evaluate how well the generated molecules satisfy the target conditions and ensure their chemical validity, we propose the following metrics:

- Mol Stable: The proportion of molecules with correct valence, serving as the most reliable indicator of unconditional molecular generation quality.

- Stable MAE: The mean absolute error (MAE) of properties, calculated only for stable molecules. By focusing on chemically valid molecules, this metric mitigates the risk of models hacking classifiers (e.g., generating invalid molecules with misleading pseudo-labels).

- Average MAE: The conventional property MAE across all generated molecules, including unstable ones, which does not account for molecular stability.

The results in Table 8 demonstrate that UniGEM have the ability to generate both condition-aligned and chemically valid molecules.

### B.3 Relationship with Denoising Pre-training

UniGEM leverages denoising diffusion generative models to learn molecular representations, drawing parallels to denoising pre-training approaches (Zaidi et al., 2022; Feng et al., 2023a; Ni et al., 2023). To better understand UniGEM, we explore its framework from the perspective of pre-training.

Unlike conventional denoising pre-training methods, UniGEM introduces several distinctions. It incorporates a broader range of noise scales during the molecular growth phase and integrates an atom prediction task. However, it does not employ the specialized noise distributions proposed by Feng et al. (2023a); Ni et al. (2023) and utilizes a different model backbone. Moreover, UniGEM jointly trains supervised property prediction with "unsupervised" generation tasks, without relying on additional unsupervised datasets, diverging from the typical pre-train-and-finetune paradigm.

An interesting question arises when comparing these two "pre-training" approaches: does denoising with multiple noise scales, in conjunction with atom type prediction, lead to better representations? To investigate this, we compare UniGEM with Frad, a denoising framework specifically designed for denoising pre-training. For a fair comparison, we aligned the network architecture by pre-training Frad with EGNN. The results, presented in Table 9, show that UniGEM outperforms Frad, despite the latter's focus on pre-training with specialized noise distributions. This outcome suggests that UniGEM may offer a promising approach to representation learning.

Table 9: Performance on the QM9 property prediction tasks.

| | # pretrain data | $\epsilon_{LUMO}$ | $\epsilon_{HOMO}$ | $\Delta\epsilon$ | Avg. |
|---|---|---|---|---|---|
| Uni-MOL (Uni-MOL) | 19M | - | - | - | 127.1 |
| Frad (TorchMD-NET) | 3.4M (PCQMv2) | 13.7 | 15.3 | 27.8 | 18.9 |
| Frad (EGNN) ) | 3.4M (PCQMv2) | 22.1 | 21.1 | 36.8 | 26.7 |
| UniGEM (EGNN) | 0.1M (QM9) | **16.7** | **20.9** | **34.5** | **24.0** |

### C  Hyperparameter Settings

The hyperparameter settings are detailed in Table 10. Our model is based on EDM (Hoogeboom et al., 2022), with two additional tunable hyperparameters introduced by our training algorithm: nucleation time and shared layers. Notably, these hyperparameters are guided by prior knowledge and do not require extensive tuning, highlighting the practical convenience of our approach.

Table 10: Network and training hyperparameters.

| Network Hyperparameters | Value |
| --- | --- |
| Embedding size | 256 |
| Layer number | 9 for QM9, 4 for Geom-Drugs |
| Shared layers | 1 |

| Training Hyperparameters | Value |
| --- | --- |
| Batch size | 64 for QM9, 32 for Geom-Drugs |
| Train epoch | 2000 for QM9, 13 for Geom-Drugs |
| Learning rate | $1.00 \times 10^{-4}$ |
| Optimizer | Adam |
| $T$ (sample times) | 1000 |
| Nucleation time | 10 |
| Oversampling ratio | 0.5 for each branch |
| Loss weight | 1 for each loss term |

The nucleation time, $t_n$, has a clear physical interpretation: it represents the moment when the molecular scaffold begins to form during the denoising process. Thus the selection of $t_n$ is not an arbitrary hyperparameter adjustment but is guided by the underlying physical principles. This parameter can be reasonably estimated from the noise schedule of the diffusion model and is largely independent of dataset size, model architecture, or optimization algorithms. Observing the noise process in EDM's coordinate denoising suggests that $t_n \in [0, 100]$ is an appropriate range where the molecular structure remains discernible.

Further, prior studies (Zaidi et al., 2022; Feng et al., 2023a) indicate that adding noise with a standard deviation of $\sigma \in [0.02, 0.05]$ preserves molecular semantics. Using EDM's noise schedule $\sigma(t)$, this corresponds to $t_n/T \approx 0.01 \sim 0.03$, which translates to $t_n = 10 \sim 30$. Our experiments further confirm that the choice of $t_n$ is robust to variations in other parameters (Figure 2). Therefore, the value of $t_n$ can be almost determined before training.

We conducted experiments with three different choices of shared layers ($k = 1$, 3, and 5) and found minimal impact on property prediction, with $k = 1$ and $k = 3$ outperforming $k = 5$ in generation.

For QM9, we trained for 2000 epochs with a batch size of 64, which took approximately 7 days on a single A100 GPU. For GEOM-drugs, we trained for 13 epochs with a batch size of 32x4=128, taking 6.5 days on four A100 GPUs.

## D    RELATED WORK

### D.1    3D MOLECULE GENERATION

The technological development of generative AI has driven numerous research efforts in 3D molecular generation. Luo & Ji (2022); Gebauer et al. (2019) use autoregressive methods to generate molecules, while Garcia Satorras et al. (2021) employs normalizing flows (Chen et al., 2018) for molecule generation. Recently, many works have applied diffusion models (Ho et al., 2020; Sohl-Dickstein et al., 2015), which have achieved success in image generation, to 3D molecular geometric generation. Hoogeboom et al. (2022) proposed an equivariant diffusion model (EDM) for 3D molecular generation. Building on this, EDM-Bridge (Wu et al., 2022) proposes adding prior bridges (Peluchetti, 2023) to improve EDM performance. GeoLDM (Xu et al., 2023) suggests using an autoencoder to encode molecules into a latent space, and then performing diffusion generation in the latent space. GeoBFN (Song et al., 2024a) introduces a new generative technique, Bayesian Flow Networks (Graves et al., 2023), to 3D molecular generation, applying Bayesian inference to the parameter space of molecular distributions.

Generating atom types in diffusion-based molecular generation remains a significant challenge due to their discrete nature. EDM represents atom types as one-hot vectors and generates them through diffusion model. In contrast, other approaches specifically design methods for discrete data gen-

eration. The BFN algorithm introduces tailored generation strategies respectively for discrete and discretized data, inspiring GeoBFN to treat atom types as atomic numbers and employ the BFN algorithm for discreted data, which incorporates a binning technique to map continuous probabilities to discrete values. Additionally, Discrete Flow Models (Campbell et al., 2024) have been proposed for discrete data generation using Continuous Time Markov Chains and have been applied to molecular generation in Lin et al. (2025). These methods generate atom types simultaneously with atomic coordinates, inevitably requiring specialized algorithms for discrete data generation. UniGEM, on the other hand, circumvents this challenge by generating only atomic coordinates and subsequently predicting atom types based on the generated structures. This approach effectively leverages molecular properties to bypass the difficulties associated with generating discrete data.

The methods following EDM adopt its evaluation framework and dataset settings. However, recently, alternative approaches to 3D molecular generation have emerged with different setups (Vignac et al., 2023; Le et al., 2024; Irwin et al., 2024). In terms of input, these methods incorporate bond information in addition to atomic types and coordinates for representing 3D molecules. Regarding evaluation, they use a bond inference strategy distinct from EDM's rules. EDM defines bonds strictly based on interatomic distances, implicitly enforcing constraints on bond lengths and steric hindrance. In contrast, these newer methods rely on model-inferred bonds without such constraints, significantly influencing stability and validity metrics(Vignac et al., 2023). Consequently, evaluating these methods requires additional energy-related metrics and geometric distribution constraints to enable a fair comparison with EDM-based methods.

### D.2 3D Molecular Pre-training

Molecular pre-training methods aim to learn molecular representations that are useful for various downstream tasks, especially property prediction. Common molecular pre-training methods include graph masking (Hu et al., 2020; Hou et al., 2022; Rong et al., 2020; Feng et al., 2023b), contrastive learning (Stärk et al., 2022; Liu et al., 2021; 2023; Li et al., 2022), and 3D denoising. Among them, 3D denoising, due to its interpretability in learning force fields, has been proven to be a very effective pre-training strategy (Zaidi et al., 2022; Feng et al., 2023a; Ni et al., 2023; 2024), especially for quantum chemical properties (Feng et al., 2024). Recently, some pre-training works have integrated diffusion as a pre-training task (Song et al., 2024b) or incorporated diffusion into the pre-training process (Liu et al., 2023; Chen et al., 2023a), demonstrating that useful molecular representations can be learned during the molecular generation process.

## E INTRODUCTION OF JOINT DIFFUSION FOR MOLECULES

A 3D molecule is represented by coordinates and atom types. Formally, we denote a molecule with $M$ atoms as $\boldsymbol{z} = (\boldsymbol{x}, \boldsymbol{h})$, where $\boldsymbol{x} = (\boldsymbol{x}_1, \cdots, \boldsymbol{x}_M) \in \mathbb{R}^{3M}$ represents the atomic positions, and $\boldsymbol{h} = (\boldsymbol{h}_1, \cdots, \boldsymbol{h}_M) \in \{\boldsymbol{e}_0, \cdots, \boldsymbol{e}_H\}^M$ encodes the atom type information. Each $\boldsymbol{h}_i$ is a one-hot vector of dimension $H$, corresponding to the type of the atom $i$, where $H$ is the total number of distinct atom types in the dataset.

Recently, several works have applied diffusion models (Sohl-Dickstein et al., 2015; Ho et al., 2020) to 3D molecular data, employing joint diffusion to simultaneously generate molecular coordinates and atom types (Hoogeboom et al., 2022; Guan et al., 2023; Gao et al., 2024). The diffusion model independently injects noise to the coordinates and atom types respectively via a forward process:

$$q(\boldsymbol{z}_t|\boldsymbol{z}_0) = \mathcal{N}_x(\boldsymbol{x}_t|\alpha_t \boldsymbol{x}_0, \sigma_t^2 \boldsymbol{I}) \cdot \mathcal{N}(\boldsymbol{h}_t|\alpha_t \boldsymbol{h}_0, \sigma_t^2 \boldsymbol{I}), \quad t = 1, \cdots, T \tag{10}$$

where $z_0$ is a molecule in the dataset, $\alpha_t$ and $\sigma_t$ represent the noise schedule and satisfy $\alpha_t^2 + \sigma_t^2 = 1$ monotonically decrease from 1 to 0. $\mathcal{N}_x$ represents the Gaussian distribution in the zero center-of-mass (CoM) subspace satisfying $\sum_i^M \boldsymbol{x}_i = \boldsymbol{0}$. $\cdot$ refers to joint distribution.

To generate samples, the forward process is reversed using a Markov chain $r_\theta(\boldsymbol{z}_{0:T}) = r_\theta(\boldsymbol{z}_T) \prod_{t=1}^T r_\theta(\boldsymbol{z}_{t-1}|\boldsymbol{z}_t)$ with a noise term approximated by a neural network $\phi_\theta^{(h)}(\boldsymbol{z}_t, t)$:

$$r_\theta(\boldsymbol{z}_{t-1}|\boldsymbol{z}_t) = \mathcal{N}_x\left(\boldsymbol{x}_{t-1}|\tilde{\boldsymbol{\mu}}\left(\boldsymbol{x}_t, \phi_\theta^{(x)}(\boldsymbol{z}_t, t)\right), \tilde{\sigma}^2 \boldsymbol{I}\right) \cdot \mathcal{N}\left(\boldsymbol{h}_{t-1}|\tilde{\boldsymbol{\mu}}\left(\boldsymbol{h}_t, \phi_\theta^{(h)}(\boldsymbol{z}_t, t)\right), \tilde{\sigma}^2 \boldsymbol{I}\right) \tag{11}$$

where $\tilde{\boldsymbol{\mu}}\left(i_t, \boldsymbol{\phi}_\theta^{(i)}(\boldsymbol{z}_t, t)\right) = \frac{1}{\alpha_{t|t-1}}i_t - \frac{\sigma_{t|t-1}^2}{\alpha_{t|t-1}\sigma_t}\boldsymbol{\phi}_\theta^{(i)}(\boldsymbol{z}_t, t)$, for $i \in \{\boldsymbol{x}, \boldsymbol{h}\}$, $\tilde{\sigma} = \sigma_{t|t-1}\sigma_{t-1}/\sigma_t$, $\alpha_{t|t-1} = \alpha_t/\alpha_{t-1}$ and $\sigma_{t|t-1}^2 = \sigma_t^2 - \alpha_{t|t-1}^2\sigma_{t-1}^2$.

The prior distribution is approximated by $p(\boldsymbol{z}_N) = \mathcal{N}_x(\boldsymbol{x}_N|\boldsymbol{0}, \boldsymbol{I}) \cdot \mathcal{N}(\boldsymbol{h}_N|\boldsymbol{0}, \boldsymbol{I})$. The noise prediction network is trained using a mean squared error (MSE) loss $\min_\theta \mathbb{E}_{t \sim U(0,T]}\mathcal{L}_t$:

$$\mathcal{L}_t = \mathbb{E}_{q(\boldsymbol{z}_0, \boldsymbol{z}_n)}\|\boldsymbol{\phi}_\theta(\boldsymbol{z}_t, t) - \boldsymbol{\epsilon}_t\|^2 \simeq \mathbb{E}_{q(\boldsymbol{z}_0, \boldsymbol{z}_n)}\|\boldsymbol{\phi}_\theta(\boldsymbol{z}_t, t) - (-\sigma_t\nabla_{\boldsymbol{z}_t}\log q(\boldsymbol{z}_t))\|^2, \quad (12)$$

where $\epsilon_t = (\boldsymbol{z}_t - \alpha_t\boldsymbol{z}_0)/\sigma_t$ is the standard Gaussian noise injected to $\boldsymbol{z}_0$ and $\nabla_{\boldsymbol{z}_t}\log q(\boldsymbol{z}_t))$ refers to the score function. The equivalent loss is derived using the equivalence between score matching and conditional score matching (Vincent, 2011b), with a formal proof provided in Ni et al. (2024); Feng et al. (2023a). Thus $\mathcal{L}_t$ is termed denoising loss or equivalently score estimation loss. Loss in equation 12 can be further decomposed as atom type and coordinate noise prediction: $\mathcal{L}_t = \mathbb{E}_{q(\boldsymbol{z}_0, \boldsymbol{z}_n)}\|\boldsymbol{\phi}_\theta^{(x)}(\boldsymbol{z}_t, t) - \boldsymbol{\epsilon}_t^{(x)}\|^2 + \|\boldsymbol{\phi}_\theta^{(h)}(\boldsymbol{z}_t, t) - \boldsymbol{\epsilon}_t^{(h)}\|^2 := \mathcal{L}_t^{(x|h)} + \mathcal{L}_t^{(h|x)}$.

## F    REPRESENTATION LEARNING ANALYSIS FOR MOLECULAR COORDINATE GENERATION

We analyze from the perspective of information maximization (InfoMax) (Linsker, 1988; Oord et al., 2018) the reasons why training with denoising diffusion loss can effectively learn molecular representations that enhance property prediction, as well as why property prediction is more effective after ready time.

### F.1    INFOMAX TARGET

In accordance with the principles of information maximization (InfoMax), our objective is to select a representation $\zeta_t$ that maximizes the mutual information (MI) between the input data and its representation. In the context of UniGEM, this objective translates to maximizing the mutual information between the original molecular coordinates $\boldsymbol{x}_0$ and the learned latent representations $\zeta_t$. Here, $\zeta_t$ is derived from the intermediate layers of the denoising network, with the input consisting of the noisy coordinates $\boldsymbol{x}_t$ at time step $t$. This latent representation $\zeta_t$ is subsequently utilized in the denoising task. A graphical representation of this model is illustrated in Figure 6.

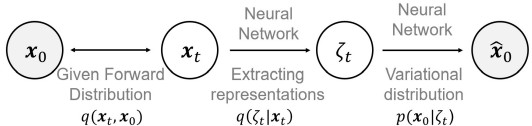

Figure 6: Graphical model of representation learning in the diffusion model.

We define $q(\boldsymbol{x}_0, \boldsymbol{x}_t)$ as the given forward distribution, while $q(\zeta_t|\boldsymbol{x}_t)$ and $p(\boldsymbol{x}_0|\zeta_t)$ represent the network-modeled latent representation distribution and the denoising predictive distribution, respectively. We denote $q(\boldsymbol{x}_0, \boldsymbol{x}_t, \zeta_t) := q(\boldsymbol{x}_0, \boldsymbol{x}_t)q(\zeta_t|\boldsymbol{x}_t)$, and $q(\boldsymbol{x}_0|\zeta_t) := \frac{\int q(\boldsymbol{x}_0, \boldsymbol{x}_t, \zeta_t)d\boldsymbol{x}_t}{\int q(\boldsymbol{x}_0, \boldsymbol{x}_t, \zeta_t)d\boldsymbol{x}_t d\boldsymbol{x}_0}$. The mutual information target to be maximized can be expressed as follows:

$$
\begin{aligned}
I(\zeta_t, \boldsymbol{x}_0) &= \mathbb{E}_{q(\boldsymbol{x}_0, \boldsymbol{x}_t, \zeta_t)}\left[\log\frac{q(\boldsymbol{x}_0|\zeta_t)}{q(\boldsymbol{x}_0)}\right] \\
&= H(\boldsymbol{x}_0) - \mathbb{E}_{q(\boldsymbol{x}_0, \boldsymbol{x}_t, \zeta_t)}\left[\log\frac{q(\boldsymbol{x}_0|\boldsymbol{x}_t)}{q(\boldsymbol{x}_0|\zeta_t)}\right] + \mathbb{E}_{q(\boldsymbol{x}_0, \boldsymbol{x}_t)}\left[\log q(\boldsymbol{x}_0|\boldsymbol{x}_t)\right] \\
&= H(\boldsymbol{x}_0) - H(\boldsymbol{x}_0|\boldsymbol{x}_t) - \mathbb{E}_{q(\boldsymbol{x}_0, \boldsymbol{x}_t, \zeta_t)}\left[\log\frac{q(\boldsymbol{x}_0|\boldsymbol{x}_t)}{q(\boldsymbol{x}_0|\zeta_t)}\right] \\
&= I(\boldsymbol{x}_0; \boldsymbol{x}_t) - \mathbb{E}_{q(\boldsymbol{x}_t, \zeta_t)}\left[D_{\mathrm{KL}}(q(\boldsymbol{x}_0|\boldsymbol{x}_t)\|q(\boldsymbol{x}_0|\zeta_t))\right]
\end{aligned}
\quad (13)
$$

This expression reveals that in order to learn a good representation $\zeta_t$, it is essential to maximize the term $I(\boldsymbol{x}_0; \boldsymbol{x}_t)$ while minimizing the Kullback-Leibler (KL) divergence term.

- The first term, $I(\boldsymbol{x}_0; \boldsymbol{x}_t)$, quantifies the information retained between the original and noisy molecular coordinates and is monotonically decreasing with respect to time $t$. Given that the KL divergence is nonnegative, $I(\boldsymbol{x}_0; \boldsymbol{x}_t)$ serves as the upper bound for $I(\zeta_t, \boldsymbol{x}_0)$, a result known as the data processing inequality. Therefore, at later stages of the generative process, such as when $t \leq t_r$, $I(\zeta_t, \boldsymbol{x}_0)$ can become larger, particularly when the second KL divergence term is minimized.

- The second term, however, is challenging to minimize since $q(\boldsymbol{x}_0|\zeta_t)$ is intractable. To address this, we employ a variational approach to derive a tractable upper bound for the KL divergence term. Furthermore, we will demonstrate how this KL divergence term can be minimized within the denoising task, thereby validating the representation learning capability of the denoising process.

### F.2 VARIATIONAL UPPER BOUND ON THE KL DIVERGENCE TERM

Since $\mathbb{E}_{q(\boldsymbol{x}_t, \zeta_t)}\left[D_{\mathrm{KL}}(q(\boldsymbol{x}_0|\zeta_t)\|p(\boldsymbol{x}_0|\zeta_t))\right] \geq 0$, the following inequality holds:

$$\int q(\boldsymbol{x}_0|\zeta_t) \log q(\boldsymbol{x}_0|\zeta_t)\, d\boldsymbol{x}_0 \geq \int q(\boldsymbol{x}_0|\zeta_t) \log p(\boldsymbol{x}_0|\zeta_t)\, d\boldsymbol{x}_0 \tag{14}$$

Thus, we obtain the following expression:

$$\mathbb{E}_{q(\boldsymbol{x}_0, \boldsymbol{x}_t, \zeta_t)}\left[\log \frac{q(\boldsymbol{x}_0|\boldsymbol{x}_t)}{q(\boldsymbol{x}_0|\zeta_t)}\right] \leq \mathbb{E}_{q(\boldsymbol{x}_0, \boldsymbol{x}_t, \zeta_t)} \log \frac{q(\boldsymbol{x}_0|\boldsymbol{x}_t)}{p(\boldsymbol{x}_0|\zeta_t)} = \mathbb{E}_{q(\boldsymbol{x}_t, \zeta_t)}\left[D_{\mathrm{KL}}(q(\boldsymbol{x}_0|\boldsymbol{x}_t)\|p(\boldsymbol{x}_0|\zeta_t))\right] \tag{15}$$

Consequently, by minimizing this variational upper bound in the right hand side of equation 15, we can effectively reduce the KL divergence term in equation 13.

### F.3 RELATION TO DENOISING LOSS

In this section, we discuss the relationship between $\mathbb{E}_{q(\boldsymbol{x}_t, \zeta_t)}\left[D_{\mathrm{KL}}(q(\boldsymbol{x}_0|\boldsymbol{x}_t)\|p(\boldsymbol{x}_0|\zeta_t))\right]$ and the denoising loss. Given that $q(\boldsymbol{x}_t|\boldsymbol{x}_0) = \mathcal{N}_x(\alpha_t \boldsymbol{x}_0, \sigma_t^2 \boldsymbol{I})$, we can apply Tweedie's formula (Efron, 2011; Luo, 2022) to obtain $\mathbb{E}_q[\boldsymbol{x}_0|\boldsymbol{x}_t] = \frac{1}{\alpha_t}(\boldsymbol{x}_t + \sigma_t^2 \nabla \log q(\boldsymbol{x}_t))$. Please note that while the distribution is Gaussian in the zero center-of-mass (CoM) subspace, this does not alter the form of Tweedie's formula. A detailed derivation is provided below:

$$\begin{aligned}
\mathbb{E}_q[\boldsymbol{x}_0|\boldsymbol{x}_t] &= \int \boldsymbol{x}_0 q(\boldsymbol{x}_0|\boldsymbol{x}_t) d\boldsymbol{x}_0 = \int \boldsymbol{x}_0 \frac{q(\boldsymbol{x}_t|\boldsymbol{x}_0)q(\boldsymbol{x}_0)}{q(\boldsymbol{x}_t)} d\boldsymbol{x}_0 \\
&= \frac{\int \boldsymbol{x}_0 \frac{1}{(2\pi\sigma_t^2)^{3(M-1)/2}} \exp(-\frac{1}{2\sigma_t^2}||\boldsymbol{x}_t - \alpha_t \boldsymbol{x}_0||^2) q(\boldsymbol{x}_0) d\boldsymbol{x}_0}{q(\boldsymbol{x}_t)} \\
&= \frac{\int [(\alpha_t \boldsymbol{x}_0 - \boldsymbol{x}_t) + \boldsymbol{x}_t]\frac{1}{(2\pi\sigma_t^2)^{3(M-1)/2}} \exp(-\frac{1}{2\sigma_t^2}||\boldsymbol{x}_t - \alpha_t \boldsymbol{x}_0||^2) q(\boldsymbol{x}_0) d\boldsymbol{x}_0}{\alpha_t q(\boldsymbol{x}_t)} \\
&= \frac{\int \sigma_t^2 \nabla_{\boldsymbol{x}_t} q(\boldsymbol{x}_t|\boldsymbol{x}_0) q(\boldsymbol{x}_0) d\boldsymbol{x}_0}{\alpha_t q(\boldsymbol{x}_t)} + \frac{\int \boldsymbol{x}_t q(\boldsymbol{x}_0|\boldsymbol{x}_t) d\boldsymbol{x}_0}{\alpha_t} \\
&= \frac{\sigma_t^2 \nabla_{\boldsymbol{x}_t} \int q(\boldsymbol{x}_t|\boldsymbol{x}_0) q(\boldsymbol{x}_0) d\boldsymbol{x}_0}{\alpha_t q(\boldsymbol{x}_t)} + \frac{\boldsymbol{x}_t}{\alpha_t} = \frac{\sigma_t^2 \nabla_{\boldsymbol{x}_t} \log q(\boldsymbol{x}_t) + \boldsymbol{x}_t}{\alpha_t}
\end{aligned} \tag{16}$$

Recall that $q(\zeta_{\mathbf{t}}|\mathbf{x}_t)$ and $p(\mathbf{x}_0|\zeta_{\mathbf{t}})$ represent the network-modeled distributions. In practice, our neural network does not involve random variables; rather, we define it to parametrize the expectation of $p(\mathbf{x}_0|\zeta_{\mathbf{t}})$ which means $\frac{\mathbf{x}_t - \sigma_t \phi_\theta^{(x)}(\mathbf{x}_t, t)}{\alpha_t} = \mathbb{E}_p[\mathbf{x}_0|\zeta_{\mathbf{t}}]$ and $\zeta_{\mathbf{t}}$ is a deterministic function of $\mathbf{x}_t$, denoted as $\zeta_{\mathbf{t}} = g_\theta(\mathbf{x}_t)$. Thus, the denoising diffusion loss can be rewritten as the $L_2$ distance between two expectations.

$$\begin{aligned}
\mathcal{L}_t^{(x)} &= \mathbb{E}_{q(\boldsymbol{x}_0,\boldsymbol{x}_n)}\|\phi_\theta^{(x)}(\boldsymbol{x}_t,t) - (-\sigma_t\nabla_{\boldsymbol{x}_t}\log q(\boldsymbol{x}_t|\boldsymbol{x}_0))\|^2 \\
&= \mathbb{E}_{q(\boldsymbol{x}_0,\boldsymbol{x}_n)}\|\phi_\theta^{(x)}(\boldsymbol{x}_t,t) - (-\sigma_t\nabla_{\boldsymbol{x}_t}\log q(\boldsymbol{x}_t))\|^2 + C \\
&= \mathbb{E}_{q(\boldsymbol{x}_0,\boldsymbol{x}_n)}\|\phi_\theta^{(x)}(\boldsymbol{x}_t,t) - \frac{\boldsymbol{x}_t - \alpha_t\mathbb{E}_q[\boldsymbol{x}_0|\boldsymbol{x}_t]}{\sigma_t}\|^2 + C \\
&= \mathbb{E}_{q(\boldsymbol{x}_0,\boldsymbol{x}_n)}\frac{\alpha_t^2}{\sigma_t^2}\|\mathbb{E}_q[\boldsymbol{x}_0|\boldsymbol{x}_t] - (\boldsymbol{x}_t - \sigma_t\phi_\theta^{(x)}(\boldsymbol{x}_t,t))/\alpha_t\|^2 + C \\
&= \mathbb{E}_{q(\boldsymbol{x}_0,\boldsymbol{x}_n)}\frac{\alpha_t^2}{\sigma_t^2}\|\mathbb{E}_q[\boldsymbol{x}_0|\boldsymbol{x}_t] - \mathbb{E}_p[\boldsymbol{x}_0|g_\theta(\boldsymbol{x}_t)]\|^2 + C,
\end{aligned} \tag{17}$$

where $C$ is constant independent of $\theta$ (Vincent, 2011a). It is important to note that, in general, the KL divergence cannot be upper-bounded by the $L_2$ distance between expectations for arbitrary distributions. However, in the case of Gaussian distributions with identical variance, these two quantities can be equivalent. Specifically, when both $p(\boldsymbol{x}_0|\zeta_t)$ and $q(\boldsymbol{x}_0|\boldsymbol{x}_t)$ follow Gaussian distributions with the same variance $\sigma$, the following relationship holds:

$$\begin{aligned}
\mathbb{E}_{q(\boldsymbol{x}_t,\zeta_t)}\left[D_{\mathrm{KL}}(q(\boldsymbol{x}_0|\boldsymbol{x}_t)\|p(\boldsymbol{x}_0|\zeta_t))\right] &= \mathbb{E}_{q(\boldsymbol{x}_t,\zeta_t)}\left[\frac{1}{2\sigma^2}\|\mathbb{E}_q[\boldsymbol{x}_0|\boldsymbol{x}_t] - \mathbb{E}_p[\boldsymbol{x}_0|\zeta_t]\|^2\right] \\
&= \frac{1}{2\sigma^2}\mathbb{E}_{q(\boldsymbol{x}_0,\boldsymbol{x}_n)}\|\mathbb{E}_q[\boldsymbol{x}_0|\boldsymbol{x}_t] - \mathbb{E}_p[\boldsymbol{x}_0|g_\theta(\boldsymbol{x}_t)]\|^2 = \frac{\sigma_t^2}{2\sigma^2\alpha_t^2}\mathcal{L}_t,
\end{aligned} \tag{18}$$

The distribution $q(\boldsymbol{x}_0|\boldsymbol{x}_t)$ is typically approximated as Gaussian in empirical Bayes methods, and a similar Gaussian approximation is employed during the reverse process in diffusion models. We assume that $p(\boldsymbol{x}_0|\zeta_t)$ also follows a Gaussian distribution with the same variance as $q(\boldsymbol{x}_0|\boldsymbol{x}_t)$.

Thus, minimizing the denoising loss corresponds to minimizing the KL divergence term, which contributes to optimizing the lower bound of the InfoMax objective in equation 13.

### F.4 THE BENEFITS OF DIFFUSION FOR PROPERTY PREDICTION: A FORCE LEARNING PERSPECTIVE

The denoising task, particularly at specific time steps during diffusion training, has been shown to effectively learn meaningful molecular representations, as demonstrated in prior work (Zaidi et al., 2022; Feng et al., 2023a; Ni et al., 2024; Arts et al., 2023). This process can be interpreted as approximating atomic forces, with the accuracy of these forces being highly sensitive to the noise scale applied during training (Feng et al., 2023a; Ni et al., 2024). Specifically, the noise scale must strike a balance—neither too large nor too trivial—in order to yield pre-trained representations that are beneficial for molecular property prediction. Thus, the denoising training steps in the growth phase may capture valuable molecular features by implicitly learning the atomic forces.

### G  GENERATIVE ERROR ANALYSIS OF UNIGEM AND JOINT DIFFUSION MODEL

#### G.1  GENERAL DIFFUSION ERROR BOUND

We follow (Chen et al., 2023b) to derive the error bound for diffusion model and make the following mild assumptions on the data distribution $q(y_0)$. To distinguish it from the notation of molecules, we use $y$ to represent general data generated by the diffusion model.

**Assumption 1 (Lipschitz score).** For all $t \geq 0$, the score $\nabla \ln q(y_t)$ is $L_y$-Lipschitz.

**Assumption 2 (second moment bound).** For some $\eta > 0$, $\mathbb{E}_{q(y_0)}[\|Y_0\|^{2+\eta}]$ is finite. Denote $m_y^2 = \mathbb{E}_{q(y_0)}[\|Y_0\|^2]$ for the second moment of $q(y_0)$.

$\tilde{T}$ refers to the timestep of the simple Ornstein-Uhlenbeck (OU) process as defined by Chen, described by the equation

$$d\bar{Y}_t = -\bar{Y}_t \, dt + \sqrt{2} \, dB_t, \quad t \in [0, \tilde{T}],$$

while our forward process is given by

$$d\bar{Y}_t = -g(t)^2 \bar{Y}_t \, dt + \sqrt{2} \, g(t) \, dB_t, \quad t \in [0, T],$$

where g(t) is determined by our noise schedule $\alpha_t$. The timesteps of the two processes differ by a time reparameterization. In EDM, $T$ also refers to the discretization steps of the diffusion process.

**Theorem G.1** (Theorem2 in Chen et al. (2023b)). *Suppose that Assumptions 1 and 2 hold. Suppose that the step size $l := \tilde{T}/T$ satisfies $l \lesssim 1/L$, where $L \geq 1$. Then, it holds that*

$$TV(p_\theta(\boldsymbol{y}_0), q(\boldsymbol{y}_0)) \lesssim \underbrace{\sqrt{KL(q(\boldsymbol{y}_0)||p_\theta(\boldsymbol{y}_T))} \exp(-\tilde{T})}_{\text{prior distribution error}} + \underbrace{(L_y \sqrt{d_y l} + L_y m_y l)\sqrt{\tilde{T}}}_{\text{discretization error}} + \underbrace{\sqrt{l \sum_{t=1}^{T} \mathcal{L}_t}}_{\text{score estimation error}},$$

(19)

*where $f_1 \lesssim f_2$ denotes that there is a universal constant $C$ such that $f_1 \leq f_2$, $d_y$ is the dimensionality of data y.*

In Theorem G.1, we modified the last term compared to original paper. The original formulation is $\sqrt{\max_t\{\mathcal{L}_t\}\tilde{T}}$, but we adopted a tighter expression used in the proof of Theorem 10 in Chen et al. (2023b), considering the average error across all timesteps: $\sqrt{\frac{\tilde{T}}{T} \sum_{t=1}^{T} \mathcal{L}_t}$. This adjustment facilitates a more precise comparison of the error differences between the UniGEM method and the joint generation approach.

## G.2 ERROR BOUND FOR UNIGEM AND JOINT DIFFUSION

Based on the results above, we can derive the molecular generation error bound for UniGEM and Joint Diffusion.

**Theorem 2.2-1.** With Assumptions 1 and 2, the total variation between the generated data distribution and ground-truth data distribution is bounded by:

$$TV(r_\theta(\boldsymbol{z}_0), q(\boldsymbol{z}_0)) \lesssim \underbrace{\sqrt{KL(q(\boldsymbol{z}_0)||r_\theta(\boldsymbol{z}_T))} \exp(-\tilde{T})}_{\text{prior distribution error}} + \underbrace{(L_z \sqrt{d_z l} + L_z m_z l)\sqrt{\tilde{T}}}_{\text{discretization error}}$$

$$+ \underbrace{\sqrt{l \sum_{t=1}^{T} \mathcal{L}_t^{(x|h)}}}_{\text{coordinate score estimation error}} + \underbrace{\sqrt{l \sum_{t=1}^{T} \mathcal{L}_t^{(h|x)}}}_{\text{atom type score estimation error}},$$

(20)

where $q(\boldsymbol{z}_T) = \mathcal{N}_{xh}(\alpha_t \boldsymbol{z}_0, \boldsymbol{I})$, with $\alpha_t$ being a small value close to zero, $\mathcal{N}_{xh}$ represents the joint distribution of the corresponding subspace Gaussian of $\boldsymbol{x}$ and the Gaussian of $\boldsymbol{h}$, and $m_z^2 = \mathbb{E}_{q(\boldsymbol{z}_0)} \|\boldsymbol{z}_0\|^2$ is the second moment of $q(\boldsymbol{z}_0)$.

*Proof.* This result is largely a direct application of the previous theorem, with two key distinctions. Firstly, instead of operating in a standard Gaussian distribution in $3M$ dimensional space, the molecular coordinates have to be projected into the zero center of mass (CoM) subspace. This projection introduces a constant factor difference in the probability density. Specifically, the distribution of the projected standard Gaussian in the zero CoM subspace is given by $p(\boldsymbol{x}) = \mathcal{N}_x(\boldsymbol{x}|\boldsymbol{0}, \boldsymbol{I}) = \frac{1}{(2\pi)^{3(M-1)/2}} \exp(-\frac{1}{2}\|\boldsymbol{x}\|^2) = (2\pi)^{3/2}\mathcal{N}(\boldsymbol{0}, \boldsymbol{I})$, where $\boldsymbol{x} \in \mathbb{R}^{3M}$ and has zero CoM. In this case, the bound still holds, with further details available in You et al. (2023).

Secondly, the score estimation error can be further decomposed. We begin by decomposing the joint score function as follows:

$$\nabla_{\boldsymbol{z}_t} \log q(\boldsymbol{z}_t) = (\nabla_{\boldsymbol{x}_t} \log q(\boldsymbol{z}_t), \nabla_{\boldsymbol{h}_t} \log q(\boldsymbol{z}_t)) = (\nabla_{\boldsymbol{x}_t} \log q(\boldsymbol{x}_t|\boldsymbol{h}_t), \nabla_{\boldsymbol{h}_t} \log q(\boldsymbol{h}_t|\boldsymbol{x}_t)) \quad (21)$$

In the forward process $x_t$ and $h_t$ are conditional independent given $x_0$ and $h_0$. Therefore, we can express the conditional probability as $q(\boldsymbol{x}_t|\boldsymbol{h}_t) = \int q(\boldsymbol{x}_t|\boldsymbol{h}_t, \boldsymbol{x}_0, \boldsymbol{h}_0)q(\boldsymbol{x}_0, \boldsymbol{h}_0)d\boldsymbol{x}_0 d\boldsymbol{h}_0 = \int q(\boldsymbol{x}_t|\boldsymbol{x}_0, \boldsymbol{h}_0)q(\boldsymbol{x}_0, \boldsymbol{h}_0)d\boldsymbol{x}_0 d\boldsymbol{h}_0 = \int q(\boldsymbol{x}_t|\boldsymbol{x}_0)(\int q(\boldsymbol{x}_0, \boldsymbol{h}_0)d\boldsymbol{h}_0)d\boldsymbol{x}_0 = q(\boldsymbol{x}_t)$. Thus, we obtain

$$\nabla_{\boldsymbol{z}_t} \log q(\boldsymbol{z}_t) = (\nabla_{\boldsymbol{x}_t} \log q(\boldsymbol{x}_t), \nabla_{\boldsymbol{h}_t} \log q(\boldsymbol{h}_t)) \quad (22)$$

Consequently, the score estimation error can be decomposed into the following components:

$$
\begin{aligned}
\mathcal{L}_t :=& \mathbb{E}_{q(\boldsymbol{z}_0, \boldsymbol{z}_n)} \| \boldsymbol{\phi}_\theta(\boldsymbol{z}_t, t) - (-\sigma_t \nabla_{\boldsymbol{z}_t} \log q(\boldsymbol{z}_t)) \|^2 \\
=& \mathbb{E}_{q(\boldsymbol{z}_0, \boldsymbol{z}_n)} (\| \boldsymbol{\phi}_\theta^{(x)}(\boldsymbol{z}_t, t) - (-\sigma_t \nabla_{\boldsymbol{x}_t} \log q(\boldsymbol{x}_t)) \|^2 + \| \boldsymbol{\phi}_\theta^{(h)}(\boldsymbol{z}_t, t) - (-\sigma_t \nabla_{\boldsymbol{h}_t} \log q(\boldsymbol{h}_t)) \|^2) \\
:=& \mathcal{L}_t^{(x|h)} + \mathcal{L}_t^{(h|x)}
\end{aligned}
\tag{23}
$$

As a result, we can now establish the proof of Theorem 2.2.

$$
\begin{aligned}
& TV(r_\theta(\boldsymbol{z}_0), q(\boldsymbol{z}_0)) \\
& \lesssim \sqrt{KL(q(\boldsymbol{z}_0)\|r_\theta(\boldsymbol{z}_T))} \exp(-\tilde{T}) + (L_z \sqrt{d_z l} + L_z m_z l) \sqrt{\tilde{T}} + \sqrt{l \sum_{t=1}^{T} (\mathcal{L}_t^{(h|x)} + \mathcal{L}_t^{(x|h)}))} \\
& \lesssim \sqrt{KL(q(\boldsymbol{z}_0)\|r_\theta(\boldsymbol{z}_T))} \exp(-\tilde{T}) + (L_z \sqrt{d_z l} + L_z m_z l) \sqrt{\tilde{T}} + \sqrt{l \sum_{t=1}^{T} \mathcal{L}_t^{(h|x)}} + \sqrt{l \sum_{t=1}^{T} \mathcal{L}_t^{(x|h)}}
\end{aligned}
\tag{24}
$$

$\square$

**Theorem 2.2-2.** Denote the reverse diffusion process of $\boldsymbol{x}$ is $p_\theta(\boldsymbol{x}_{0:T}) = p_\theta(\boldsymbol{x}_T) \prod_{t=1}^{T} p_\theta(\boldsymbol{x}_{t-1}|\boldsymbol{x}_t)$. The atom types are predicted by a prediction network $\boldsymbol{\phi}_\theta^{(h)}(\boldsymbol{x}_0) := \boldsymbol{\phi}_\theta^{(h)}(\boldsymbol{x}_0, t = 0) = (p_\theta(\boldsymbol{h}_{0,1}|\boldsymbol{x}_0), \cdots, p_\theta(\boldsymbol{h}_{0,M}|\boldsymbol{x}_0))$. With Assumptions 1 and 2, the total variation between the generated data distribution and ground-truth data distribution is bounded by:

$$
TV(p_\theta(\boldsymbol{z}_0), q(\boldsymbol{z}_0)) \lesssim \underbrace{\sqrt{KL(q(\boldsymbol{x}_0)\|p_\theta(\boldsymbol{x}_T))} \exp(-\tilde{T})}_{\text{prior distribution error}} + \underbrace{(L_x \sqrt{d_x l} + L m_x l) \sqrt{\tilde{T}}}_{\text{discretization error}}
$$

$$
+ \underbrace{\sqrt{l \sum_{t=1}^{T} \mathcal{L}_t^{(x)}}}_{\text{coordinate score estimation error}} + \underbrace{\frac{1}{2} \mathbb{E}_{q(\boldsymbol{x}_0)} \mathcal{L}^{(h)}(\boldsymbol{x}_0)}_{\text{atom type estimation error}},
\tag{25}
$$

where $\mathcal{L}^{(h)}(\boldsymbol{x}_0) = |\boldsymbol{\phi}_\theta^{(h)}(\boldsymbol{x}_0) - f^{(h)}(\boldsymbol{x}_0)|$ is the atom type prediction error, with $f^{(h)}$ providing the ground truth atom type $i$ corresponding to $\boldsymbol{x}_0$ in the dataset.

*Proof.*

$$
\begin{aligned}
TV(p_\theta(\boldsymbol{z}_0), q(\boldsymbol{z}_0)) &= \frac{1}{2} \int | p_\theta(\boldsymbol{h}_0, \boldsymbol{x}_0) - q(\boldsymbol{h}_0, \boldsymbol{x}_0) | \, d\boldsymbol{x}_0 \, d\boldsymbol{h}_0 \\
&= \frac{1}{2} \int | p_\theta(\boldsymbol{h}_0|\boldsymbol{x}_0) p_\theta(\boldsymbol{x}_0) - q(\boldsymbol{h}_0|\boldsymbol{x}_0) q(\boldsymbol{x}_0) | \, d\boldsymbol{x}_0 \, d\boldsymbol{h}_0 \\
&\leq \frac{1}{2} \int | p_\theta(\boldsymbol{h}_0|\boldsymbol{x}_0) p_\theta(\boldsymbol{x}_0) - p_\theta(\boldsymbol{h}_0|\boldsymbol{x}_0) q(\boldsymbol{x}_0) | \, d\boldsymbol{x}_0 \, d\boldsymbol{h}_0 \\
&\quad + \frac{1}{2} \int | p_\theta(\boldsymbol{h}_0|\boldsymbol{x}_0) q(\boldsymbol{x}_0) - \delta_{(f^{(h)}(\boldsymbol{x}_0))} q(\boldsymbol{x}_0) | \, d\boldsymbol{x}_0 \, d\boldsymbol{h}_0 \\
&= \frac{1}{2} \int ( \int p_\theta(\boldsymbol{h}_0|\boldsymbol{x}_0) \, d\boldsymbol{h}_0) | p_\theta(\boldsymbol{x}_0) - q(\boldsymbol{x}_0) | \, d\boldsymbol{x}_0 \\
&\quad + \frac{1}{2} \mathbb{E}_{q(\boldsymbol{x}_0)} \sum_{j=1}^{M} \sum_{i=1}^{H} | p_\theta(\boldsymbol{h}_{0,j} = \boldsymbol{e}_i|\boldsymbol{x}_0) - \delta_{(f_j^{(h)}(\boldsymbol{x}_0) = \boldsymbol{e}_i)} | \\
&= TV(p_\theta(\boldsymbol{x}_0), q(\boldsymbol{x}_0)) + \frac{1}{2} \mathbb{E}_{q(\boldsymbol{x}_0)} \mathcal{L}^{(h)}(\boldsymbol{x}_0)
\end{aligned}
\tag{26}
$$

Then by using Theorem G.1 to provide the upper bound of the diffusion error of $x$, we can obtain equation 25 $\square$

**Remark of the deterministic mapping $f$.** In the datasets involved in this paper, the coordinates of different molecules are distinct. This ensures that the mapping $f$ from coordinates to atom types is well-defined. We posit that this notion of a well-defined mapping also applies to the organic molecules of interest. Consequently, there exists $f$ such that for all potential coordinates $\boldsymbol{x}_0$, $q(\boldsymbol{h}_0|\boldsymbol{x}_0) = \delta_{(f^{(h)}(\boldsymbol{x}_0))}$, justifying our architectural choice of utilizing predictive methods to determine atom types.

### G.3 GENERATIVE ERROR COMPARISON

We compare the error bounds for the two methods by analyzing each component in the upper bounds:

**a. Prior distribution error and discretization error**: These errors are influenced by the dimensionality of the generated data. Since UniGEM operates on a lower-dimensional space i.e. generating coordinates only, its error in these terms is smaller.

1. The dimensionality of generated data is smaller for UniGEM: $d_x < d_z$.

2. Since the prior distribution of $\boldsymbol{x}$ and $\boldsymbol{h}$ are standard Gaussian distributions in the reverse generative process, we have $r_\theta(\boldsymbol{z}_T) = r_\theta(\boldsymbol{x}_T)r_\theta(\boldsymbol{h}_T)$, $p_\theta(\boldsymbol{x}_T) = r_\theta(\boldsymbol{x}_T)$. Then $KL(q(\boldsymbol{z}_0)||r_\theta(\boldsymbol{z}_T)) = KL(q(\boldsymbol{x}_0, \boldsymbol{h}_0)||r_\theta(\boldsymbol{x}_T)r_\theta(\boldsymbol{h}_T)) = KL(q(\boldsymbol{x}_0)||r_\theta(\boldsymbol{x}_T)) + \mathbb{E}_{q(\boldsymbol{x}_0)}KL(q(\boldsymbol{h}_0|\boldsymbol{x}_0)||r_\theta(\boldsymbol{h}_T))$. Thus $KL(q(\boldsymbol{x}_0)||p_\theta(\boldsymbol{x}_T)) \leq KL(q(\boldsymbol{z}_0)||r_\theta(\boldsymbol{z}_T))$ and $KL(q(\boldsymbol{z}_0)||r_\theta(\boldsymbol{z}_T))$ scales with $d_z$.

3. The second moment term satisfies $m_z^2 = \mathbb{E}_{q(\boldsymbol{z}_0)}||\boldsymbol{z}_0||^2 = \mathbb{E}_{q(\boldsymbol{z}_0)}(||\boldsymbol{x}_0||^2 + ||\boldsymbol{h}||^2) = m_x^2 + m_h^2$, i.e. $m_x^2 \leq m_z^2$ and $m_z^2$ scales with $d_z$.

4. Since the distribution of $x$ and $h$ are independent in the forward process, the score function can be expressed as $\nabla_{\boldsymbol{z}_t} \ln q(\boldsymbol{z}_t) = (\nabla_{\boldsymbol{x}_t} \ln q(\boldsymbol{x}_t), \nabla_{\boldsymbol{h}_t} \ln q(\boldsymbol{h}_t))$. As a result, the Lipschitz constant for $x$ is bounded by the Lipschitz constant for the joint variable $z$, i.e. $L_x \leq L_z$.

5. Other variables $l$, $T$, $\tilde{T}$ are the same for the two methods.

**b. Type and coordinate estimation error**:

According to Theorems 2.2, we decompose the score estimation error into two components: atom type estimation error and coordinate estimation error, which are compared experimentally. For atom type estimation, we compare the accumulated denoising loss $\sqrt{l \sum_{t=1}^{T} \mathcal{L}_t^{(h|x)}}$ and prediction loss $\frac{1}{2}\mathbb{E}_{q(\boldsymbol{x}_0)}\mathcal{L}^{(h)}(\boldsymbol{x}_0)$. For coordinate estimation, we evaluate the results of two denoising losses $\sqrt{l \sum_{t=1}^{T} \mathcal{L}_t^{(x|h)}}$ and $\sqrt{l \sum_{t=1}^{T} \mathcal{L}_t^{(x)}}$. It is important to note that the input of the network differs in the two denoising losses, with $\mathcal{L}_t^{(x|h)}$ incorporating noisy atom type as input:

$$\mathcal{L}_t^{(x|h)} = \mathbb{E}_{q(\boldsymbol{z}_0, \boldsymbol{z}_n)}(||\boldsymbol{\phi}_\theta^{(x)}(\boldsymbol{z}_t, t) - (-\sigma_t \nabla_{\boldsymbol{x}_t} \log q(\boldsymbol{x}_t))||^2 \tag{27}$$

$$\mathcal{L}_t^{(x)} = \mathbb{E}_{q(\boldsymbol{x}_0, \boldsymbol{x}_n)}(||\boldsymbol{\phi}_\theta^{(x)}(\boldsymbol{x}_t, t) - (-\sigma_t \nabla_{\boldsymbol{x}_t} \log q(\boldsymbol{x}_t))||^2 \tag{28}$$

The detailed experimental comparison and analysis is presented in Section 2.2, showing UniGem achieves lower atom types and coordinates estimation error.

