# OpenReview forum: "UniGEM: A Unified Approach to Generation and Property Prediction for Molecules"
_ICLR.cc/2025/Conference — ICLR 2025 Poster_

### Official Review · Reviewer_5XEL · 2024-10-18

**Soundness:** 3
**Presentation:** 3
**Contribution:** 2
**Rating:** 6
**Confidence:** 4

**Summary:**

This paper proposes UniGEM, a unified approach for molecular generation and molecular property prediction. The authors carefully designed both generative and predictive objectives in a single unified framework. Interestingly, these two objectives exhibit a synergistic relationship, enhancing each other's performance.

**Strengths:**

- The problems of interest, molecular generation and property prediction, are important for real-world applications, e.g., drug discovery.

- Proposed method seems reasonable; (1) applying the predictive and atom type loss near t=0, (2) changing atom type loss to classification loss, and (3) learning only coordinates when t is far from 0.

- The findings are interesting; simultaneously learning both predictive and generative objective improve each other's performance.

**Weaknesses:**

- Insufficient baselines.

My major concern is the omitted baselines. For example, [1,2,3] also study 3D molecular generation and often outperform this method. I think such 3D molecular generation baselines should be in main tables for comparison.

---
- Marginal improvements.

Considering the omitted baselines [1,2,3], the improvements seem marginal for me. Furthermore, it seems that [3] largely outperforms this paper.

---
- Complexity of the framework.

UniGEM introduces many design choices, e.g., t_n and oversampling ratio. I think this may limit the generalizability to other datasets, i.e., require extensive hyper-parameter tuning.

---
Overall, I think the concept of this work is reasonably novel. However, due to the insufficient comparison with baselines, I lean towards rejection at this moment.

[1] Song et al., Equivariant Flow Matching with Hybrid Probability Transport for 3D Molecule Generation, NeurIPS 2023\
[2] Xu et al., Geometric-Facilitated Denoising Diffusion Model for 3D Molecule Generation, AAAI 2024\
[3] Irwin et al., Efficient 3D Molecular Generation with Flow Matching and Scale Optimal Transport, arXiv

**Questions:**

1. How much time does it take the overall training?

2. Can EGNN be replaced by recent 3D GNNs, e.g., [4], to further improve performance?

3. Can the authors provide visualizations of the denoising process for generated samples?

[4] Liu et al., Spherical Message Passing for 3D Molecular Graphs, ICLR 2022

---

> ### Author Response · Authors · 2024-11-20
> **Rebuttal by Authors**
>
> We greatly appreciate your expert feedback and your insightful concerns. Your concerns and questions are responded as follows:
>
> >  W1: Add recent baselines e.g.[1,2,3]；
> >
> > W2: The improvements seem marginal, especially comparing with [3].
>
> **Response**:
> 1. **Different Contributions**: The contributions of UniGEM and the methods in [1, 2, 3] are orthogonal, targeting distinct aspects of molecular generation. To enable a fair comparison with the key molecular generation baseline, EDM, we aligned the generative models and architectures, even though they are not the most advanced options. The non-marginal improvements over EDM (mol stable:82.0->89.8) highlight the contribution of UniGEM’s techniques for molecular generation.
> - In contrast, the methods in [1, 2, 3] focus on other dimensions, such as leveraging advanced architectures and novel generative models (e.g., transitioning from DDPM to flow matching). These approaches are complementary to UniGEM and could be integrated with its framework to further enhance overall performance.
> - To demonstrate that UniGEM can adapt to different generative algorithms and network structures, we implemented UniGEM with the advanced generative method GeoBFN[5], or advanced equivariant network GET[6]. Results are shown in Tables (last two rows) and in the response to Q2.
>
>
> | **Methods**          | Mol stable% | **Atom stable%** | Validity% | **V*U%**  |
> |----------------------|-------------|------------------|-----------|-----------|
> | **Other architectures and generative models**: |              |             |           |        |
> | EquiFM[1]            | 88.3        | 98.9         | 94.7      | 93.5  |
> | GFMDiff[2]          | 87.7    | 98.9         | 96.3      | **95.1**  |
> | **EDM Based**:   |              |             |           |        |
> | EDM                  | 82.0          | 98.7             | 91.9      | 90.7      |
> | UniGEM(with EDM) | **89.8**    | **99.0**           | **95.0**    | **93.2**  |
> | **BFN Based**:   |              |             |           |        |
> | GeoBFN[5]              | 90.9       | 99.1            | 95.3     | **93.0**  |
> | UniGEM(with BFN) | **93.7**    | **99.3**         | **97.3** | **93.0**  |
>
>
> 2. **Comparison with [1, 2]**:
>   - The results in the table above demonstrate that UniGEM can achieve SOTA performance, with non-marginal improvements.
>   - Among these metrics, Mol stable is the most rigorous metric for molecular correctness and serves as the most reliable measure of generative quality. Even with its original UniGEM using a basic generative model (DDPM) and architecture (EGNN), UniGEM still reports higher mol stable metrics compared to [1] and [2].  The results further demonstrate the contributions of UniGEM's simple yet effective generative method.
>
> 3. **Comparison with [3]**:
>
>   Notably, [3] and [1][2] (including our work and EDM) belong to fundamentally different classes of generative and evaluation methods, making direct comparisons infeasible.
>   - **Differences in Input Information**: UniGEM and its baselines only use atomic types and coordinates to represent 3D molecules. In contrast, [3] and its associated baselines [4] leverage additional bonding information, providing a distinct advantage in capturing molecular structure.
>   -  **Differences in Evaluation**: [3, 4] use a bond inference method different from EDM’s rules. EDM defines bonds strictly based on interatomic distances, enforcing constraints on bond lengths and steric hindrance. Conversely, [3, 4] rely on model-inferred bonds without such constraints. This difference in bonding definitions greatly affects the reported metrics.
>
>      To illustrate this, we re-evaluated EDM and UniGEM using OpenBabel software to infer bonds from generated atomic types and coordinates, following the approach of [4]. The results, shown in Table below, illustrate how bonding definitions affect the metrics.
> | Method          | Mol stable% | Atom stable% | Validity% |
> |------------------|-------------|--------------|-----------|
> | EDM             | 82.0        | 98.7         | 91.9      |
> | EDM + OBabel    | 96.8        | 99.8         | 98.3      |
> | UniGEM          | 89.8        | 99.0         | 95.0      |
> | UniGEM + OBabel | 98.0        | 99.9         | 98.5      |
>
>   -    **Challenges in Reproducibility**: EDM’s evaluation approach makse Mol stable more discriminative. Thus, to fairly compare with [3], it would be necessary to evaluate [3] using EDM's bond inference rules.  Unfortunately, the code for [3] is not publicly available, preventing us from isolating these variables for a more equitable comparison.
>
> [1] [2] [3] As you referred.
>
> [4]  MiDi: Mixed Graph and 3D Denoising Diffusion  for Molecule Generation, ECML PKDD 2023
>
> [5] Unified Generative Modeling of 3D Molecules with Bayesian Flow Networks, ICLR24
>
> [6] Sliced Denoising: A Physics-Informed Molecular Pre-Training Method, ICLR24

---

> ### Author Response · Authors · 2024-11-20
> **Rebuttal by Authors**
>
> > W3: Complexity of the hyper-parameter tuning.
>
> **Response**:
> 1. **Oversampling Ratio Does Not Require Tuning**
> - When employing a multi-branch network architecture, t<$t_n$ and t>$t_n$ can effectively be treated as separate sub-tasks handled by different branches.
> - The oversampling ratio primarily impacts the relative convergence rate of the two branches rather than the final results. For efficient training, we adopt a default ratio of 1:1, which is simple and effective.
> 2. **Nucleation Time ($t_n$) Does Not Require Costly Tuning**
>
> (a) Physical Meaning and Approximate Range
>
> $t_n$ has a clear physical interpretation as the point in the noise schedule when the molecular scaffold begins to form. This makes it relatively straightforward to estimate.
>
> The value of $t_n$ can be approximated based on the diffusion model’s noise schedule:
>   - For example, visualizations of the noise process in EDM’s coordinate denoising suggest that $t_n$≈0∼100 is appropriate, as the molecular structure remains discernible within this range (see Figure https://anonymous.4open.science/r/UniGEM_Rebuttal-DCBD/gen_process_2.jpeg).
>   - Prior studies ([1], [2]) show that adding noise with a standard deviation of σ≈0.02∼0.05 preserves molecular semantics. Mapping this to EDM’s noise schedule σ(t), we estimate $t_n$/T≈0.01∼0.03, corresponding to $t_n$=10∼30.
>
> (b) Robustness of nucleation time:
>
> Our experiments demonstrate that the model is robust to variations in $t_n$. For example, increasing $t_n$ up to 100 can reduce the performance of generation and property prediction but still yields results that significantly outperform the baseline EDM (Figure 2 in paper). This robustness indicates that precise tuning or costly retraining to determine $t_n$ can be unnecessary.
>
> [1] Pre-training via denoising for molecular property prediction, ICLR23
>
> [2] Fractional denoising for 3d molecular pre-training, ICML23
>
> ---
> > Q1. How much time does it take the overall training?
>
> **Response**:
> For QM9, we trained for 2000 epochs with a batch size of 64, which took approximately 7 days on a single A100 GPU. For GEOM-drugs, we trained for 13 epochs with a batch size of 32x4=128, taking 6.5 days on four A100 GPUs. Thanks for your question, we have added this information in appendix C.
>
> ---
> > Q2: Can EGNN be replaced by recent 3D GNNs?
>
> **Response**:
> Yes, we can adapt UniGEM to more powerful networks to further improve performance. To address this, we trained on the network architecture Geometric Equivariant Transformer (GET) from [1]. However, GET is more complex and requires more training time. Due to the time constraints of the rebuttal, we compared the performance of EGNN and GET with the same epoch count (300). The results for property prediction and generation are shown in the table below. The results indicate that UniGEM exhibits stronger performance on a more powerful network.
>
>  | Models(300 epoch)           | Mol stable% ↑ | Atom stable% ↑ | Validity% ↑ | V*U% ↑ |
>  |------------------|---------------|----------------|-------------|--------|
>  | UniGEM (EGNN)    | 0.7981        | 0.9825        | 0.9044      | 0.8909 |
> | UniGEM (GET)     | **0.9073**        | **0.9909**        | **0.9537**      | **0.9391** |
>
>  | Models(300 epoch)              | LUMO ↓ | HOMO ↓ | Gap ↓ |
> |---------------------|--------|--------|-------|
> | UniGEM(EGNN)              | 24.2   | 25.7   | 44.5  |
> | UniGEM (GET)        |  **21.2**   |  **24.5**   |  **41.1**  |
>
> Furthermore, we provide the loss curve during training and the Mol stable curve for generated molecules, showing that with a stronger network, the property prediction loss converges better, and the Mol stable metric also has the potential for further improvement. (Figure https://anonymous.4open.science/r/UniGEM_Rebuttal-DCBD/mol_stable_compare.jpeg and https://anonymous.4open.science/r/UniGEM_Rebuttal-DCBD/prop_compare.jpeg)
>
> [1] Sliced Denoising: A Physics-Informed Molecular Pre-Training Method, ICLR24
>
> ---
> > 3. Can the authors provide visualizations of the denoising process for generated samples?
>
> **Response**: Thank you for the suggestion. We have visualized the denoising process of UniGEM for generated molecules, as shown in Figure https://anonymous.4open.science/r/UniGEM_Rebuttal-DCBD/gen_process_1.jpeg. This figure illustrates the generative process over 1000 iterations, with the final atom types predicted at the last step.
>
> This visualization has been added to the updated version of the paper.

---

> ### Author Response · Authors · 2024-11-20
>
> We sincerely appreciate your recognition of the reasonable method and the value of our findings, as well as the constructive suggestions you provided.
>
> We have carefully addressed your concerns and have made necessary revisions in the manuscript (including adding the baselines you recommended to related works and comparing results in Table 7), with changes highlighted in $\color{blue}\textrm{blue}$.
>
> We are always ready to address any further questions you may have. We believe the updated version is of higher quality and hope you might consider a **more positive rating** to help our method reach a broader audience.

---

> ### Author Response · Authors · 2024-11-24
> **Eagerly Awaiting Your Response as Reviewer-Author Discussion Deadline Approaches**
>
> Dear Reviewer 5XEL,
>
> We sincerely appreciate the time and effort you have dedicated to reviewing our paper. We have carefully considered your feedback, responded to each of your questions, and revised our paper accordingly.
>
> As the reviewer-author discussion deadline approaches, we would like to kindly inquire if our responses have sufficiently addressed your questions and concerns. Please let us know if there are any remaining issues or areas needing further clarification. We are more than willing to make additional adjustments to ensure a thorough resolution of all points raised.
>
> Thank you once again for your time and expertise.
>
> Best regards,
>
> The Authors of UniGEM

---

> ### Author Response · Authors · 2024-11-25
> **Looking forward to Your Feedback as the Discussion Deadline Nears**
>
> Dear reviewer 5XEL,
>
> Thank you for the time and consideration you’ve invested in reviewing our paper. Your insights and questions have played a crucial role in enhancing our work. With the **discussion deadline approaching**, we are looking forward to your response to our recent replies. Your feedback is highly valued, and we greatly appreciate your attention to this matter. We’re eager to continue the discussion and gain further insights from you.
>
> Best regards,
>
> The Authors.

---

> > ### Comment · Reviewer_5XEL · 2024-11-26
> >
> > Thank the authors for the rebuttal. My concerns are addressed and I raised the score accordingly.

---

> > > ### Author Response · Authors · 2024-11-26
> > > **Thank You for Your Feedback**
> > >
> > > Dear reviewer 5XEL,
> > >
> > > Thank you very much for taking the time to review our rebuttal. We're pleased that our responses addressed your concerns. Wishing you all the best!
> > >
> > > The Authors of UniGEM.

---

### Official Review · Reviewer_Q62J · 2024-10-29

**Soundness:** 3
**Presentation:** 3
**Contribution:** 3
**Rating:** 8
**Confidence:** 3

**Summary:**

This paper proposes UniGEM, a unified method designed to effectively address both molecular generation and property prediction tasks. Experimental results and ablation studies demonstrate the effectiveness of the proposed approach.

**Strengths:**

1. The paper presents an innovative and well-reasoned approach. Denoising has recently emerged as a dominant pre-training method for molecular representation learning, with strong physical interpretability. Combining the diffusion model with representation learning is a promising direction and good first trial.

2. Decoupling atom type and coordinates during generation is an interesting approach. Together with theoretical insights, this approach shows promise in reducing generation error. This technique resembles some protein generation models, where pure chain geometry is initially learned before predicting amino acid sequences.

3. Experiments validate the efficacy of the proposed framework.

**Weaknesses:**

1. The experimental setup lacks depth. To better demonstrate the effectiveness of the proposed method, I suggest testing on a property-conditioned generation task using the QM9 dataset, a standard benchmark for generative models. This would offer a more challenging evaluation, testing both the model’s ability to learn representations related to specific conditions and its capacity to capture the underlying data distribution—goals that align very well with the proposed method’s objectives.

2. The comparisons lack recent SOTA methods. It would be valuable to compare UniGEM’s performance in prediction tasks against other pre-trained methods that also leverage denoising, such as UniMol or Frad (I understand that the model backbones differ, yet this comparison could at least shed some lights). This will test whether UniGEM’s approach (i.e., diffusion-based denoising, with larger noise levels and more fine-grained steps) offers advantages over conventional coarser denoising techniques in representation learning.

**Questions:**

No further questions.

---

> ### Author Response · Authors · 2024-11-19
> **Rebuttal by Authors**
>
> We greatly appreciate your expert feedback and your insightful concerns. Your concerns and questions are responded as follows:
>
> > W1: To better demonstrate the effectiveness of the proposed method, I suggest testing on a property-conditioned generation task using the QM9 dataset
>
> **Response**: Thank you very much for your valuable suggestion. We have supplemented two experiments on conditional generation.
>
> 1. **Conventional Property-Conditioned Generation**
>
> As per your suggestion, we used the traditional conditional generation method by taking the LUMO property as the network conditional input and removing the property prediction module in UniGEM. The experimental setup is consistent with the conditional generation of EDM. The results are as follows:
>
> | Model   | Mol Stable%↑ | Stable MAE↓ | Average MAE ↓|
> |---------|------------|------------|-------------|
> | EDM     | 77.96     | 0.5890     | 0.6056      |
> | **UniGEM**  | **86.09**     | **0.5642**     | **0.5804**      |
>
>
>
>
> - Evaluation Improvements:
>
> Existing evaluation methods typically train an auxiliary property classifier (e.g. EGNN) to generate pseudo-labels for the generated molecules, comparing these labels to the target condition to assess generation quality.
>   However, this approach has limitations: classifiers only use atom types and coordinates as input, meaning even invalid molecules can yield seemingly accurate property predictions.
> To improve evaluation rationality, we introduced three metrics:
>   1. Mol Stable %: The proportion of molecules with correct valence.
>   2. Stable MAE: The property MAE calculated only for stable molecules.
>   3. Average MAE: The property MAE across all generated molecules, including unstable ones.
> - UniGEM outperforms EDM on all three metrics, demonstrating its superior ability to generate condition-aligned and chemically valid molecules.
>
> 2. **UniGEM’s Unique Property-Conditioned Generation**
>
> To highlight the advantage of UniGEM’s joint molecule generation and property prediction capability, we devised a unique property-conditioned generation method:
> - Instead of retraining a conditional generation model, we used UniGEM’s pre-trained property prediction module as guidance during sampling. The update step for the molecular coordinates is formulated as follows:$x_{t-1}= \frac{1}{\alpha_{t|{t-1}}} x_t - \frac{\sigma^2_{t|{t-1}}}{\alpha_{t|{t-1}} \sigma_t}  \phi_\theta^{(x)}(x_t, t)  -\lambda \nabla L_{t}^{(c)} +\frac{\sigma_{t|t-1}\sigma_{t-1}}{\sigma_{t}} \epsilon$, where $L_{t}^{(c)}$ is the property prediction loss of UniGEM.
> - This approach eliminates the need for additional conditional generation training while still producing condition-aligned molecules.
>
> | Model              | LUMO   | HOMO   | Gap    | Mu     |Alpha|Cv|
> |--------------------|--------|--------|--------|--------|--------|--------|
> | EDM                | 0.606 | 0.356  | 0.665  | 1.111  |2.76| 1.101|
> | **UniGEM (Guidance)** | **0.592** | **0.233** | **0.511** | **0.805** |**2.22**| **0.873**|
>
>
> The table above shows results for four conditional properties. Without requiring retraining, UniGEM achieves performance comparable to or better than EDM trained explicitly with conditional data, demonstrating the flexibility and capability of our model.
> Thanks for your suggestion, we have added this experiment in Appendix B2 in the revised version.

---

> ### Author Response · Authors · 2024-11-19
> **Rebuttal by Authors**
>
> > W2: Add comparison with pre-trained methods e.g. Frad or UniMol.
>
> **Response**:
>
> 1. **Fairness in Comparison with Existing Pretraining Methods**
> - It is important to note that existing methods like UniMol and Frad utilize significantly larger pretraining datasets than UniGEM. They also adopt advanced equivariant networks, which have a significant impact on property prediction performance. These create inherent advantages for those methods, making a direct comparison less fair.
> - Despite this disparity, UniGEM has already demonstrated superior performance over the listed pretraining models in our experiments (Table 3 in paper), highlighting its strengths in property prediction tasks.
>
> 2. **Comparison with Frad with EGNN for Fair Evaluation**
> - We agree that a comparison with denoising-based pretraining methods like Frad is meaningful. To ensure a fair evaluation, we align the network architecture by applying Frad with EGNN as backbone.
>
> | Model              |  Pretrain Data      | LUMO  | HOMO  | Gap  | AVG    |
> |--------------------|----------------------|-------|-------|------|--------|
> | Uni-MOL (Uni-MOL)  | 19M                  | -     | -     | -    | 127.08 |
> | Frad (TorchMD-NET) | 3.4M (PCQMv2)        | 13.7  | 15.3  | 27.8 | 18.9   |
> | Frad (EGNN)        | 3.4M (PCQMv2)        | 22.1  | 21.1  | 36.8 | 26.7   |
> | UniGEM (EGNN)      | 0.1M (QM9)           | **16.7**  | **20.9**  | **34.5** | **24.0**   |
>
>  The table above shows the performance of different pre-training methods. The results show that UniGEM outperforms Frad using the same backbone, despite Frad's specialized noise distributions. This outcome suggests that denoising with multiple noise scales, in conjunction with atom type prediction may offer a promising approach to representation learning.
>   We have added this experiment in Appendix B3 in the revised version.

---

> ### Author Response · Authors · 2024-11-19
>
> We are grateful for your recognition of our contribution and novelty, as well as your valuable suggestions. We have carefully addressed your concerns and have made necessary revisions in the manuscript, with changes highlighted in $\color{blue}\textrm{blue}$. We are always ready to address any further questions you may have. We believe the updated version is of higher quality and hope you might consider a **more positive rating** to help our method reach a broader audience.

---

> ### Author Response · Authors · 2024-11-24
> **Eagerly Awaiting Your Response as Reviewer-Author Discussion Deadline Approaches**
>
> Dear Reviewer Q62J,
>
> We sincerely appreciate the time and effort you have dedicated to reviewing our paper. We have carefully considered your feedback, responded to each of your questions, and revised our paper accordingly.
>
> As the reviewer-author discussion deadline approaches, we would like to kindly inquire if our responses have sufficiently addressed your questions and concerns. Please let us know if there are any remaining issues or areas needing further clarification. We are more than willing to make additional adjustments to ensure a thorough resolution of all points raised.
>
> Thank you once again for your time and expertise.
>
> Best regards,
>
> The Authors of UniGEM

---

### Official Review · Reviewer_wNED · 2024-11-04

**Soundness:** 2
**Presentation:** 3
**Contribution:** 2
**Rating:** 6
**Confidence:** 4

**Summary:**

This paper proposes a unified model for molecular generation and property predictions. Since diffusion models require diffusing the 3D molecular conformers to a collapsed state, the molecular structures in the early time steps do not contain any meaningful structural information. As a result, the joint training of generation and property prediction can be inconsistent in the early stage. The author develops a theoretical framework to explain this issue. To solve this issue, this paper proposes a novel training paradigm that separates the diffusion time steps into two stages, which are nucleation time and growth time. The joint training would only be conducted during the growth time. Experimental results demonstrate the effectiveness of the proposed method.

**Strengths:**

(1) The presentation of the major idea is very clear. This reviewer can understand the motivation fluently through reading the introduction part. This article is well-organized and easy to follow;

(2) The theoretical analysis is well-developed and can explain the tradeoff phenomenon properly.

**Weaknesses:**

(1) It appears that the nucleation time cannot be determined prior to training the diffusion models. This could prove to be highly costly if there is a need to repeatedly retrain the diffusion models with varying hyperparameters, which may be impractical in real-world applications;

(2) While the theoretical framework is well-developed, this reviewer believes that the theoretical analysis section occupies an excessive portion of the paper. Given that the phenomenon observed by the authors is quite intuitive, experimental observations or straightforward explanations could suffice;

(3) Although the experimental results demonstrate the effectiveness of the proposed method, UniGem only marginally outperforms the baseline methods across most datasets and tasks. Furthermore, the molecular generation benchmark is not entirely convincing at this stage, as the evaluation metrics used in this benchmark fail to distinguish the capabilities of various state-of-the-art generation models. Also, in the molecular quantum property prediction benchmark tasks, UniGem only slightly surpasses the baseline models, and many quantum properties are absent from the reported benchmark results. Overall, the current empirical results do not sufficiently convince readers of the superiority of the proposed method;

(4) The novelty is a little bit limited from the algorithmic contribution side. The major contribution is splitting the diffusion training into two different stages and only conducting the joint training in the later stage. This is somehow not that significant in more general application domains.

**Questions:**

How do you determine the nucleation time before training diffusion models? Currently, it seems that the nucleation time cannot be determined before training the diffusion models. This gonna be extremely costly to select the most suitable nucleation time.

---

> ### Author Response · Authors · 2024-11-19
> **Rebuttal by Authors**
>
> We greatly appreciate your expert feedback and your insightful concerns. Your concerns and questions are responded as follows:
>
> > W1 & Q1: How to determine the hyperparameter $t_n$ (nucleation time)?
>
> **Response**:
> We acknowledge that nucleation time cannot be strictly determined prior to training. However, we emphasize that determining it does not require costly retraining of models.
>
> 1. **Physical meaning and approximate range**:
>
> Nucleation time has a clear physical meaning: it represents the moment when the molecular scaffold begins to form. This can be reasonably estimated from the noise schedule of the diffusion model and is largely unaffected by factors such as dataset size, model architecture, or optimization algorithms.
>
>   - For instance, visualizations of the noise process in EDM’s coordinate denoising indicate that $t_n$ =0∼100 is an appropriate range where the molecule structure remains discernible. As shown in the figure https://anonymous.4open.science/r/UniGEM_Rebuttal-DCBD/gen_process_2.jpeg.
>   - Additionally, prior studies ([1], [2]) show that adding noise with a standard deviation of σ≈0.02∼0.05 preserves molecular semantics. Using EDM’s noise schedule σ(t), this range corresponds to $t_n/T \approx 0.01 \sim 0.03$, i.e. $t_n = 10 \sim 30$.
>
> 2. **Robustness of nucleation time**:
>
> Our experiments demonstrate that the model is robust to variations in $t_n$. For example, increasing $t_n$ up to 100 can reduce the performance of generation and property prediction but still yields results that significantly outperform the baseline EDM (Figure 2 in paper). This robustness indicates that precise tuning or costly retraining to determine $t_n$ is unnecessary.
>
> [1] Pre-training via denoising for molecular property prediction, ICLR23
>
> [2] Fractional denoising for 3d molecular pre-training, ICML23
>
> ---
>
> > W2: The theoretical section is excessively long, as the experiments already provide intuitive explanations that seem sufficient.
>
> **Response**:
> We appreciate the reviewer’s recognition of the intuitive explanation of our findings. However, we believe the theoretical framework provides valuable insights that cannot be fully attained through experiments alone. For example,
>   - Experiments show that joint training in the later stages is superior to joint training throughout all stages. Intuitive explanations suggest that molecules are less noisy in the later stages, but they lack precision and depth.
>   - Our theory supplements two things: 1. Why diffusion training enables molecular representation learning, offering a clear foundation for its effectiveness. 2. Representation quality improves monotonically over time in diffusion, providing a precise explanation that aligns with experimental findings.
>
> We fully understand your consideration of the length. In order to better balance the content of the paper without losing key insights, we have compressed the theoretical content in the main text by **more than half a page** in the revised version.

---

> ### Author Response · Authors · 2024-11-19
> **Rebuttal by Authors**
>
> > W3.1: UniGem only marginally outperforms the baseline methods across most datasets and tasks.
>
> **Response**:
> Firstly, regarding the results for the generation task in Table 1: since our method is a modification based on EDM, the most fair comparison should be between EDM (line 3) and UniGEM. It can be seen that our method shows a significant improvement in generation metrics compared to EDM. Other baselines, such as EDM-Bridge and GeoLDM, have introduced enhancements to the generation algorithm itself, while we retained the same DDPM diffusion generation algorithm as EDM.
>
> As UniGEM is a flexible framework that can adapt to various generation algorithms, we expect it to produce even better results when adapted to more advanced generation algorithms. To illustrate this, we conducted additional experiments with UniGEM using the current SOTA generation algorithm, BFN[3,4]. Specifically, we replaced EDM's coordinate generation schedule with BFN’s coordinate generation algorithm. The results are shown in the table below, indicating that UniGEM can still achieve a significant improvement on top of a stronger generation algorithm.
> | **Methods**          | Mol stable% | **Atom stable%** | Validity% | **V*U%**  |
> |----------------------|-------------|------------------|-----------|-----------|
> | **Other generation algorithms**: |              |             |           |        |
> | EDM-Bridge           | 84.6        | 98.8             | 92.0        | 90.7      |
> | GeoLDM               | 89.4        | 98.9             | 93.8      | 92.7      |
> | **EDM Based**:   |              |             |           |        |
> | EDM                  | 82.0          | 98.7             | 91.9      | 90.7      |
> | **UniGEM(with EDM)** | **89.8**    | **99.0**           | **95.0**    | **93.2**  |
> | **BFN Based**:   |              |             |           |        |
> | GeoBFN[3]              | 90.9       | 99.1            | 95.3     | **93.0**  |
> | **UniGEM(with BFN)** | **93.7**    | **99.3**         | **97.3** | **93.0**  |
>
> Secondly, for the property prediction results in Table 2 in paper, since our method does not introduce any additional pre-training data or self-supervised strategy techniques, a fair comparison should be made with EGNN trained from scratch (line 1, using the same training data and network structure). Our UniGEM demonstrates a significant improvement in property prediction in this fair comparison.
>
> ---
> > W3.2: Evaluation metrics used in this benchmark fail to distinguish the capabilities of various state-of-the-art generation models.
>
> **Response**: Firstly, the metrics we selected include atom stability, molecule stability, validity, and valid & unique, all of which have been used by previous unconditional generation algorithms as standards for evaluating algorithm superiority.
>
> Secondly, among these metrics, Molecule stability is the most rigorous metric for molecular correctness and serves as the most reliable measure of generative quality. Because by definition, validity and atom stability are almost always greater than mol stability. So we should focus more on this metric to distinguish model capabilities.
>
> Finally, in a fair comparison using the same coordinate generation scheduling algorithm, our method demonstrates a significant improvement over EDM in molecule stability (89.8 vs. 82.0) and validity(95.0 vs 91.9). These metrics effectively highlight the strengths of different methods.
>
> ---
> > W3.3: Many quantum properties are absent from the reported benchmark results.
>
> **Response**: Thank you for your suggestion. Due to the limited time for the rebuttal, we have added results for four additional properties: U, U0, G, and H, as shown in the table below. Compared to the baseline (EGNN trained from scratch), our method further improves the performance of these properties, consistent with the findings reported in the paper.
> | Task | $G$ | $H$ | $U$ | $U_0$|
> | ---- | ---- | ---- | ---- | ---- |
> | EGNN(train from scratch) | 12.0 | 12.0 | 12.0 | 11.0|
> | UniGEM | **8.9** | **9.2** | **9.7** | **9.1**|

---

> ### Author Response · Authors · 2024-11-19
> **Rebuttal by Authors**
>
> > W4: Concerns about algorithmic novelty and significance in other domains.
>
> **Response**:
>
> 1. Algorithmic contributions:
>
>   -  Our first contribution is the integration of generation and prediction tasks, two critical tasks for drug discovery. Notably, our approach is the first to demonstrate that these tasks can mutually enhance each other and the first to unify them within a single model.
>   - We would like to emphasize the second algorithmic contribution you may overlooked: By leveraging diffusion for continuous coordinate generation and classification for discrete atom types, we address the challenge of generating discrete and continuous variables simultaneously. This is a recognized problem in molecular generation [3], and our solution has been positively received by other reviewers. For example:
>     - Reviewer Q62J found the method “interesting” and “promising.”
>     - Reviewer 5XEL described it as “reasonable.”
>   - Experimental results in the previous rebuttal table also show that our simple decoupled approach outperforms GeoBFN[3], which specifically designs a generative algorithm for discrete variables and another for continuous variables.
>
> 2. Broader applicability in AI for Science:
> - Our method is highly generalizable across molecular domains, including drug-like molecules, crystals, and organic materials, which all require generating coordinates and *atom types*.
> - The approach extends to larger biomolecules like proteins, RNA, and peptides, where generating both continuous coordinates and discrete *residue types* is critical. Reviewer Q62J also noted its potential for protein generation as a promising direction.
>
> [3] Unified Generative Modeling of 3D Molecules with Bayesian Flow Networks, ICLR24
>
> [4] Bayesian flow networks, 2023

---

> ### Author Response · Authors · 2024-11-19
>
> We sincerely appreciate your recognition of our main idea and your valuable suggestions.
>
> We have carefully addressed your concerns and have made necessary revisions in the manuscript, with changes highlighted in $\color{blue}\textrm{blue}$.
>
> We are always ready to address any further questions you may have. We believe the updated version is of higher quality and hope you might consider a **more positive rating** to help our method reach a broader audience.

---

> ### Author Response · Authors · 2024-11-24
> **Eagerly Awaiting Your Response as Reviewer-Author Discussion Deadline Approaches**
>
> Dear reviewer wNED,
>
> We sincerely appreciate the time and effort you have dedicated to reviewing our paper. We have carefully considered your feedback, responded to each of your questions, and revised our paper accordingly.
>
> As the reviewer-author discussion deadline approaches, we would like to kindly inquire if our responses have sufficiently addressed your questions and concerns. Please let us know if there are any remaining issues or areas needing further clarification. We are more than willing to make additional adjustments to ensure a thorough resolution of all points raised.
>
> Thank you once again for your time and expertise.
>
> Best regards,
>
> The Authors of UniGEM.

---

> ### Author Response · Authors · 2024-11-25
> **Looking forward to Your Feedback as the Discussion Deadline Nears**
>
> Dear Reviewer wNED,
>
> We are deeply grateful for your invaluable suggestions to help improve our paper. We kindly ask if there are any remaining issues that need further attention to meet your expectations and potentially enhance the overall assessment. Your time and feedback are greatly appreciated, and we look forward to your response.
>
> Best regards,

---

### Meta-Review · Area_Chair_Kmka · 2024-12-20

**Metareview:**

The paper proposes a novel pretraining approach for molecule property prediction and generation that aims to better align two main types of pretraining, namely prediction-based and generation-base.

Reviewers found the proposal of integration of generation and prediction based pretraining as diffusion with deep supervision novel and promising.

The rebuttal phase has cleared all significant issues Reviewers have raised. All reviewers voted ultimately for accepting the paper. Perhaps most importantly, the rebuttal phase has cleared that the proposed approach allows achieving state of the art results, as the original version of the manuscript omitted certain baselines.

A remaining concern is that improvements remain somewhat marginal in certain benchmarks. However, the novelty of the approach warrants optimism about the potential impact of the paper.

All in all, it is my pleasure to recommend acceptance of the paper. Please make sure to address reviewers’ remarks in the camera ready version.

**Additional Comments On Reviewer Discussion:**

The authors clarified the novelty of their unified framework, expanded benchmarks, provided detailed ablation studies, and formalized nucleation time selection, effectively addressing a large portion of the raised issues.

---

### Decision · Program_Chairs · 2025-01-22

Accept (Poster)